# LIMSSR: LLM-Driven Sequence-to-Score Reasoning under Training-Time Incomplete Multimodal Observations

**Huangbiao Xu** [1 2] **Huanqi Wu** [1 2] **Xiao Ke** [1 2] **Yuxin Peng** [3]

## Abstract

Real-world multimodal learning is often hindered by missing modalities. While Incomplete Multimodal Learning (IML) has gained traction, existing methods typically rely on the unrealistic assumption of full-modal availability during training to provide reconstruction supervision or cross-modal priors. This paper tackles the more challenging setting of IML under *training-time incomplete observations*, which precludes reliance on a "God's eye view" of complete data. We propose **LIMSSR** (**L**LM-Driven **I**ncomplete **M**ultimodal **S**equence-to-**S**core **R**easoning), a framework that reformulates this challenge as a conditional sequence reasoning task. LIMSSR leverages the semantic reasoning capabilities of Large Language Models via *Prompt-Guided Context-Aware Modality Imputation* and *Multi-dimensional Representation Fusion* to infer latent semantics from available contexts without direct reconstruction. To mitigate hallucinations, we introduce a *Mask-Aware Dual-Path Aggregation* to dynamically calibrate inference uncertainty. Extensive experiments on three Action Quality Assessment datasets demonstrate that LIMSSR significantly outperforms state-of-the-art baselines without relying on complete training data, establishing a new paradigm for data-efficient multimodal learning. Code is available at https://github.com/XuHuangbiao/LIMSSR.

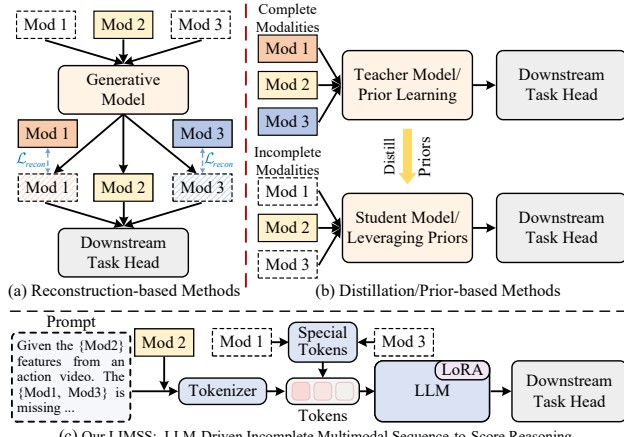

*Figure 1.* (a) Reconstruction-based and (b) Distillation/Prior-based incomplete multimodal learning methods typically rely on the unrealistic "full-modal training" assumption to generate missing data or distill priors. In contrast, (c) our **LIMSSR** leverages the context-aware reasoning capabilities of LLMs to infer latent semantics from available modalities under the more challenging *Training-Time Incomplete Observations*, eliminating the dependency on complete training data or reconstruction supervision.

## 1. Introduction

The success of multimodal learning in perceptual tasks largely hinges on the complementarity of multi-source information. However, a significant "Reality Gap" exists between ideal laboratory settings and real-world deployment, where missing modalities are inevitable due to sensor failures, privacy shielding, or data corruption. Consequently, Incomplete Multimodal Learning (IML) has emerged as a critical paradigm, requiring models to function robustly on arbitrary subsets of available modalities without catastrophic performance degradation. This capability is particularly vital and challenging in domains such as multimodal emotion recognition (Xu et al., 2024d; McKinzie et al., 2023), medical analysis (Meng et al., 2024), action recognition (Woo et al., 2023), and action quality assessment (Xu et al., 2026).

Current IML approaches generally fall into two categories: *Reconstruction-based* methods (Yoon et al., 2018; Wen et al., 2024) (Figure 1(a)), which generate missing data or latent spaces from available inputs, and *Distillation/Prior-based*

---

[1]Fujian Provincial Key Laboratory of Networking Computing and Intelligent Information Processing, College of Computer and Data Science, Fuzhou University, Fuzhou 350108, China [2]Engineering Research Center of Big Data Intelligence, Ministry of Education, Fuzhou 350108, China [3]Wangxuan Institute of Computer Technology, Peking University, Beijing 100871, China. Correspondence to: Xiao Ke <kex@fzu.edu.cn>, Yuxin Peng <pengyuxin@pku.edu.cn>.

*Proceedings of the 43rd International Conference on Machine Learning*, Seoul, South Korea. PMLR 306, 2026. Copyright 2026 by the author(s).

methods (Li et al., 2024; Xu et al., 2024d) (Figure 1(b)), which employ complete modalities as teachers to distill knowledge or learn joint priors. A common premise of these methods is the assumption of "Full-modal training," where complete data is available during training to provide supervision or priors. This assumption, however, often contradicts reality. Due to collection costs (Liu et al., 2021) or privacy constraints (Jaiswal & Provost, 2020), training data often suffers from systemic missingness. This paper addresses this gap: learning robust representations under *training-time incomplete observations* without relying on a "God's eye view" of complete modalities. The core challenge lies in the lack of a mechanism to "imagine" or complete missing semantics without direct supervision from complete data.

To address this, we propose a shift in perspective, formulating incomplete multimodal learning as a conditional sequence reasoning problem, as depicted in Figure 1(c). Given partially observed modalities and a missingness mask, the model must infer useful latent representations for the missing parts, and aggregate multimodal information for downstream prediction. We argue that Large Language Models (LLMs) are uniquely suited as the "reasoning engine" for this task. Beyond being sequence models, LLMs possess extensive world knowledge and reasoning capabilities (Feng et al., 2026b; Li et al., 2025a; Yuxin et al., 2026; Xiao et al., 2026). By explicitly injecting missingness structures and task requirements via prompts, LLMs can leverage the context of available modalities and built-in priors to infer the *Latent Semantics* of missing parts, rather than relying on accurate pixel-level reconstruction. Furthermore, the tokenized sequence nature of LLMs allows us to model "missingness" explicitly as special tokens, converting "imputation" into a standard next-token prediction or latent representation generation task. Our goal is not to force the LLM to "understand" the specific downstream task from scratch, but to utilize its sequence reasoning and conditional mechanisms to construct a generalizable framework for missing modality imputation and fusion.

To achieve this, we propose **LIMSSR** (short for **L**LM-Driven **I**ncomplete **M**ultimodal **S**equence-to-**S**core **R**easoning), a novel framework leveraging the sequence understanding and generative capabilities of LLMs for incomplete multimodal learning. Specifically, on the input side, we propose **P**rompt-Guided **C**ontext-Aware **M**odality **I**mputation (**PCMI**). We organize modality blocks with specific boundary tokens and represent missing modalities with distinct missing tokens to form equal-length placeholder sequences, prompting the LLM to infer hidden states based on the available context. Then, on the interface side, we introduce **L**LM-Driven **M**ultidimensional **R**epresentation **F**usion (**LMRF**). Instead of disrupting the LLM's raw output, LMRF employ specific fusion tokens as slots to carry multidimensional fused representations, encouraging the

LLM to blend multimodal information into latent representations aligned with the downstream task. To prevent sequence modeling collapse and hallucinations caused by missing data, we finally propose **M**ask-Aware **D**ual-Path **A**ggregation (**MDA**) on the output side. This module converts missingness into mask vectors and employs mask-aware gating to refine latent representations, dynamically balancing high-level contextual reasoning with low-level cross-modal feature aggregation.

Recognizing that task loss alone is insufficient to constrain the latent space effectively, we introduce specific optimization strategies. We employ *Consistency Learning* to constrain the consistency between the dual paths, promoting accuracy in both representation semantics and latent missing reasoning. Furthermore, we implement *Token-level Metric Regularization* to force different fusion tokens to learn distinct feature dimensions, preventing feature collapse.

We evaluate our approach in the context of long-term Action Quality Assessment (AQA) under incomplete multimodal settings. AQA is an emerging and challenging task with high stability requirements (e.g., sports scoring and medical evaluation) that must operate efficiently under various modality combinations. Extensive experiments demonstrate the effectiveness of our method on three public AQA datasets: FS1000 (Xia et al., 2023), Fis-V (Xu et al., 2019), and Rhythmic Gymnastics (Zeng et al., 2020). Our contributions are summarized as follows:

- This paper novelly formulates Incomplete Multimodal AQA (IMAQA) as a conditional sequence-to-score reasoning task and systematically investigates the training-time incomplete multimodal learning where modalities are missing during both training and testing. This breaks the dependency on complete training data and better aligns with real-world data acquisition scenarios.

- We propose LIMSSR, a novel LLM-driven incomplete multimodal sequence-to-score reasoning framework that employs specific tokens to guide LLMs in latent representation completion and fusion reasoning, and incorporates a mask-aware aggregation mechanism to resolve reasoning uncertainty.

- We validate the missing information reasoning capability of LLMs. Our method outperforms existing supervision-dependent methods even under training-time incomplete multimodal observations, achieving the new state-of-the-art on three public benchmarks.

## 2. Related Work

**Multimodal Action Quality Assessment.** With the advancement of computer vision technology (Li et al., 2026a;b; 2025e;d), the focus on fine-grained tasks has been growing.

AQA evaluates the quality of fine-grained action execution and has garnered significant attention for its applications in sports scoring (Ke et al., 2024; Xu et al., 2024b;c; 2025d; Zhou et al., 2025), rehabilitation (Bruce et al., 2024), and skill determination (Xu et al., 2025b). Multimodal learning (Lai et al., 2025; Cai et al., 2024; 2025; Wu et al., 2025; Huang et al., 2025; Feng et al., 2026a; Liu et al., 2024; 2025b; Lin et al., 2026b;a; Liu et al., 2026) has made significant progress across various tasks by leveraging effective auxiliary information. Similarly, due to the high semantic abstraction and fine-grained understanding required than traditional recognition and detection tasks (Feng et al., 2023; Ke et al., 2025; Yang et al., 2025), many AQA works enhance performance by supplementing core visual data with auxiliary modalities such as audio (Xia et al., 2023), optical flow (Zeng & Zheng, 2024), or skeletons (Bruce et al., 2024). Recently, large vision-language models (Xu et al., 2024a; 2025a;c) and LLMs (Dibenedetto et al., 2025) have been explored to leverage textual information for AQA.

However, these methods typically assume full modality availability during both training and testing, overlooking real-world modality missingness. To address this, MCMoE (Xu et al., 2026) introduces Incomplete Multimodal AQA (IMAQA), utilizing lightweight completion and mixture-of-experts (Li et al., 2025b; 2026c) to handle missing data. *Distinct from prior work, we address the more challenging training-time incomplete observations. We formulate IMAQA as a conditional sequence reasoning task, exploiting the semantic reasoning capabilities of LLMs to achieve modality imputation and fusion without reliance on full-modal supervision.*

**Incomplete Multimodal Learning.** IML is gaining increasing attention, as multimodal models often suffer from performance degradation when modalities are missing—a frequent real-world issue (Wu et al., 2024). Existing IML methods generally fall into two categories. The first is reconstruction-based methods, such as ActionMAE (Woo et al., 2023), IMDer (Wang et al., 2023), Gain (Yoon et al., 2018), and DMVG (Wen et al., 2024), which reconstruct missing modalities representation via generative or probabilistic models. The second is distillation/prior-based methods. Methods like CorrKD (Li et al., 2024) and PCD (Chen et al., 2024) introduce knowledge distillation to alleviate inconsistencies caused by missing modalities. In contrast, GCNet (Lian et al., 2023), MoMKE (Xu et al., 2024d) and MCMoE (Xu et al., 2026) rely on joint representation priors to compensate for the absence of modalities. Recently, the strong generalization ability of large language models has made them a promising paradigm for IML. Several studies (Xu et al., 2025e; Shi et al., 2024; Lee et al., 2023; Guo et al., 2024) adopt prompt-learning strategies to handle missing modalities. MissRAG (Pipoli et al., 2025) proposes a retrieval-augmented generation (RAG) framework to com-

pensate for missing modalities by retrieving semantically relevant modality tokens from a prototype pool. TAMML (Tsai et al., 2025) unifies heterogeneous and unseen modalities by converting all inputs into a shared textual space.

However, these approaches typically rely on complete training data or external resources to provide reconstruction supervision, distillation/prior targets, or to activate LLMs, which is often infeasible in real-world scenarios. *In contrast, our method operates without the requirement for complete modalities during training, instead leveraging the reasoning capabilities of LLMs to infer the latent semantics of missing modalities without direct supervision.*

## 3. Approach

We outline the preliminaries and the overview of LIMSSR, followed by detailed introductions of each module. The overall framework is illustrated in Figure 2.

### 3.1. Preliminaries

**Standard Incomplete Multimodal Learning (IML).** In standard IML scenarios, the training set is typically assumed to be complete, denoted as $\mathcal{D}_{full} = \{(\mathbf{X}_i, y_i)\}_{i=1}^N$, where $\mathbf{X}_i = \{\mathbf{X}_i^1, \ldots, \mathbf{X}_i^M\}$ encompasses all $M$ modalities. The objective is to learn a function $F$ that approximates the ground truth $y$ during inference using only a subset $\tilde{\mathbf{X}} \subset \mathbf{X}$. Most existing methods hinge on the availability of complete data during training to construct reconstruction losses $\mathcal{L}_{rec}(\mathbf{X}, \hat{\mathbf{X}})$ or distillation targets $\mathcal{L}_{distill}(F(\mathbf{X}), F(\tilde{\mathbf{X}}))$ to guide the learning process.

**Training-Time Incomplete Observations.** In this work, we address a more rigorous and realistic setting where the "God's eye view" of complete data is absent. We define the training set itself as incomplete: $\mathcal{D}_{train} = \{(\mathbf{X}_i \odot \boldsymbol{m}_i, \boldsymbol{m}_i, y_i)\}_{i=1}^N$. Here, $\boldsymbol{m}_i \in \{0, 1\}^M$ is the missingness mask for the $i$-th sample. If the $j$-th element $\boldsymbol{m}_{i,j} = 0$, the $j$-th modality $\mathbf{X}_i^j$ is unavailable not only as input but also as a ground truth target for reconstruction or distillation. Our goal is to learn a robust mapping $\mathcal{F} : (\mathcal{X}_{obs}, \boldsymbol{m}) \to y$ relying solely on partial observations $\mathcal{X}_{obs}$ and prior knowledge, without direct supervision from complete modalities.

**Instantiation: Action Quality Assessment (AQA).** We instantiate this problem within the context of multimodal long-term AQA. The input space consists of video ($\mathbf{X}^v$), audio ($\mathbf{X}^a$), and optical flow ($\mathbf{X}^f$) feature matrices, and the output $y \in \mathbb{R}$ is the action quality score. This task involves long-term temporal dependencies, high semantic abstraction, and fine-grained detail understanding, serving as an ideal testbed for evaluating the reasoning capabilities of Large Language Models (LLMs).

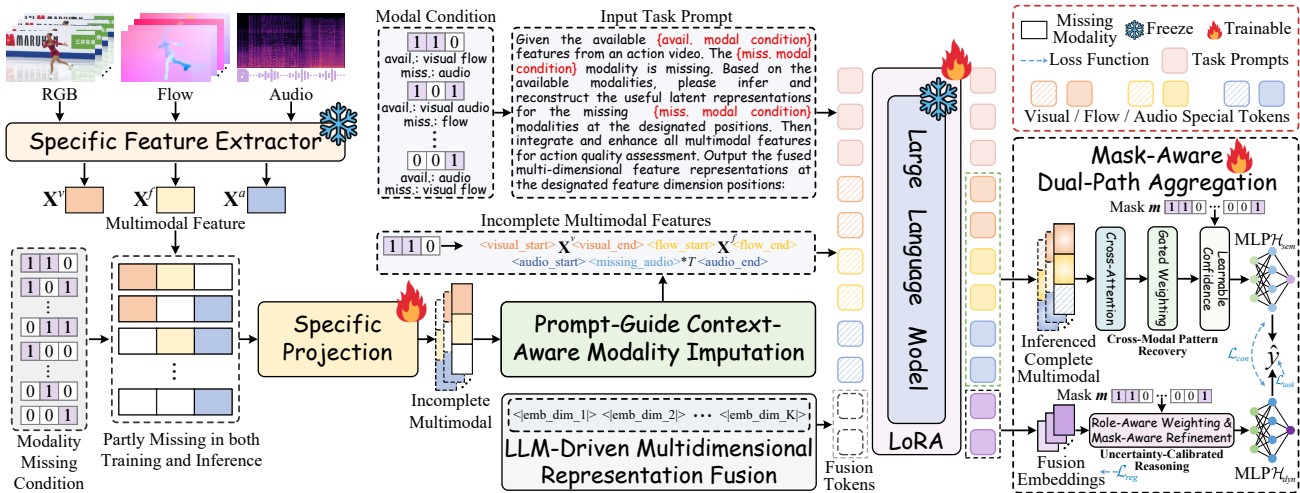

*Figure 2.* The overall framework of LIMSSR. Given incomplete data, LIMSSR first performs Prompt-Guided Context-Aware Modality Imputation, transforming available modalities into token sequences while using placeholders for missing ones. Guided by structured prompts, an LLM infers latent semantics for missing parts and consolidates multimodal information into fusion tokens. Finally, Mask-Aware Dual-Path Aggregation decodes these representations via uncertainty-calibrated reasoning and cross-modal pattern recovery paths, dynamically weighting them based on missingness states to generate robust scores.

### 3.2. Overview

As illustrated in Figure 2, we formulate Incomplete Multimodal AQA (IMAQA) as a **Conditional Sequence Reasoning** problem. LIMSSR creates a mapping $\mathcal{F}_\theta$ : $(\mathcal{X}_{obs}, \boldsymbol{m}) \to \hat{y}$ through three core phases:

1. **Context Construction ($\Phi_{in}$):** Task-related text prompts $P_{inst}$, available heterogeneous modal features $\mathcal{X}_{obs}$, special missing tokens $\mathbf{E}_{miss}$ and special fusion tokens $\mathbf{E}_{fusion}$ are unified into a latent embedding space:

$$\mathbf{Z}_{in} = \Phi_{in}(P_{inst}, \mathcal{X}_{obs}, \mathbf{E}_{miss}, \mathbf{E}_{fusion}). \quad (1)$$

2. **Latent Inference (LLM):** The LLM acts as a reasoning engine to infer missing semantics and fuse information conditioned on the available context $\mathbf{Z}_{in}$:

$$\mathbf{H}_{out} = \text{LLM}(\mathbf{Z}_{in}). \quad (2)$$

3. **Dual-Path Aggregation ($\Psi_{agg}$):** The output hidden states $\mathbf{H}_{out}$ are decoded via a mask-aware dual-path mechanism $\Psi_{agg}$ in conjunction with the missing mask vector $\boldsymbol{m}$ to generate the final predicted score $\hat{y}$:

$$\hat{y} = \Psi_{agg}(\mathbf{H}_{out}, \boldsymbol{m}). \quad (3)$$

### 3.3. Feature Extraction and Projection

Following the standard long-term AQA, we utilize frozen pre-trained backbones (Liu et al., 2022; Ke et al., 2026; Feng et al., 2026c) to extract segment-level spatiotemporal features (Li et al., 2025c). For Rhythmic Gymnastics and Fis-V datasets, the input videos are divided into $T$ non-overlapping 32-frame segments $\mathbf{I}_T$. We employ VST (Liu

et al., 2022), AST (Gong et al., 2021), and I3D (Carreira & Zisserman, 2017) to extract RGB ($\mathbf{X}^v \in \mathbb{R}^{T \times d_v}$), Audio ($\mathbf{X}^a \in \mathbb{R}^{T \times d_a}$), and Flow ($\mathbf{X}^f \in \mathbb{R}^{T \times d_f}$) features, respectively. For the FS1000 dataset, we follow prior works by using 5-second segments with a 3-second overlap, and extract multimodal features using TimeSformer (Bertasius et al., 2021), AST, and I3D. Formally,

$$\mathbf{X}^m = \text{Extractor}_m(\mathbf{I}_T), \quad m \in \{v, a, f\} \quad (4)$$

where $d_m$ denotes the feature dimension for modality $m$.

To align these multimodal features with the LLM's input space, we introduce a learnable projection block $\phi_m$ : $\mathbb{R}^{d_m} \to \mathbb{R}^{D_{LLM}}$ for each modality $m$. This block consists of two convolutional layers with ReLU non-linearity, resulting in aligned feature sequences $\tilde{\mathbf{X}}^m = \phi_m(\mathbf{X}^m)$.

### 3.4. Prompt-Guided Modality Imputation

Unlike traditional methods that zero-out missing data, we propose **Prompt-Guided Context-Aware Modality Imputation (PCMI)** to treat missingness as a variable to be inferred using prompt instruction $P_{inst}$ and placeholders.

**Modal Sequence Construction.** For each modality $m$, we design two special tokens `<m_start>` and `<m_end>` to denote the start and end of its semantic representation. This serves to separate modality information from the prompt. For an observed modality $m \in \mathcal{M}_{obs}$ ($\boldsymbol{m}_j = 1$), we construct a sequence block $\mathbf{S}^m$ with explicit semantic boundaries:

$$\mathbf{S}^m = [\text{<m\_start>}, \tilde{\mathbf{X}}^m, \text{<m\_end>}]. \quad (5)$$

For a missing modality $m \in \mathcal{M}_{miss}$ ($\boldsymbol{m}_j = 0$), we construct

a placeholder sequence:

$$\mathbf{S}^m = [\texttt{<m\_start>}, \mathbf{E}^m_{miss}, \texttt{<m\_end>}], \quad (6)$$

where $\mathcal{M}_{obs} \cup \mathcal{M}_{miss} = \{v, a, f\}, \mathcal{M}_{obs} \cap \mathcal{M}_{miss} = \emptyset$, $\mathbf{E}^m_{miss} \in \mathbb{R}^{T \times D_{LLM}}$ consists of $T$ repeated repetitions of a learnable embedding corresponding to a specific token $\texttt{<missing\_m>}$. This informs the LLM that the data is not simply "empty," but is a latent variable to be reasoned about.

**Prompt-Based Contextual Reasoning.** To leverage the LLM's conditional reasoning, we design task-specific prompts $P_{inst}$ that explicitly describe the task state: "*Given the available {avail. modal condition} features from an action video. The {miss. modal condition} modality is missing. Based on the available modalities, please infer and reconstruct the useful latent representations for the missing {miss. modal condition} modalities at the designated positions.*" The prompt guides the LLM to infer missing semantics from available context. Here, {modal condition} denotes the current combination of available and missing modalities specified by the mask vector $\boldsymbol{m}$. The final input is $\mathbf{Z}_{in} = [P_{inst}, \{\mathbf{S}^m | m \in \mathcal{M}_{obs}\}, \{\mathbf{S}^m | m \in \mathcal{M}_{miss}\}]$. Through LLM reasoning, the hidden states at missing positions $\mathbf{H}^m_{miss}$ are contextually inferred:

$$\mathbf{H}^m_{miss} = \text{LLM}(\mathbf{Z}_{in})\big|_{\text{positions of } \mathbf{E}^m_{miss}}. \quad (7)$$

### 3.5. LLM-Driven Representation Fusion

**Fusion Token Strategy.** To preserve the LLM's long-sequence generation capability, we avoid traditional pooling. Instead, we propose **LLM-Driven Multidimensional Representation Fusion (LMRF)** via special fusion tokens. Specifically, we append $K$ fusion tokens $\mathbf{E}_{fusion} = [\texttt{<emb\_dim\_1>}, \dots, \texttt{<emb\_dim\_K>}]$ to the prompt. These tokens act as "information slots" that aggregate task-relevant features from the preceding multimodal context.

Then, we revise the input task prompt to explicitly instruct the LLM to integrate multimodal information into these fusion tokens: "*Then integrate and enhance all multimodal features for action quality assessment. Output the fused multi-dimensional feature representations at the designated feature dimension positions:*". We append this to the original prompt $P_{inst}$, and denote the new complete task prompt as $P'_{inst}$. Thus, the input becomes $\mathbf{Z}_{in} = [P'_{inst}, \{\mathbf{S}^m | m \in \mathcal{M}_{obs}\}, \{\mathbf{S}^m | m \in \mathcal{M}_{miss}\}, \mathbf{E}_{fusion}]$, and we extract the last-layer outputs at these positions: $\mathbf{H}_{fusion} = \{\boldsymbol{h}_1, \dots, \boldsymbol{h}_K\} \in \mathbb{R}^{K \times D_{LLM}}$.

**Role-Aware Weighted Aggregation.** Assuming different token embeddings capture distinct evaluation dimensions (e.g., action difficulty, execution quality, and artistic expression), we define learned role weights $\boldsymbol{w}_{role} \in \mathbb{R}^K$ to

compute the main fused representation vector $\boldsymbol{z}_{main}$:

$$\boldsymbol{z}_{main} = \sum_{k=1}^{K} \text{Softmax}(\boldsymbol{w}_{role})_k \cdot \boldsymbol{h}_k. \quad (8)$$

This encourages the LLM to compress diverse evaluation criteria into specific token slots.

### 3.6. Mask-Aware Dual-Path Aggregation

Pure LLM inference (Path 1) risks hallucinations under severe missingness, while statistical feature aggregation (Path 2) lacks high-level semantics. To address this, we design **Mask-Aware Dual-Path Aggregation (MDA)**, combining both via the mask vector $\boldsymbol{m}$ to enhance robustness.

**Path 1: Uncertainty-Calibrated Reasoning.** We employ a conditional gating mechanism to calibrate the LLM's reasoning uncertainty. We compute a confidence gate $\boldsymbol{g}$ and a residual correction term $\boldsymbol{\delta}$:

$$\boldsymbol{g} = \sigma(\text{MLP}_{gate}([\boldsymbol{z}_{main}, \boldsymbol{m}])), \quad (9)$$
$$\boldsymbol{\delta} = \text{MLP}_{res}([\boldsymbol{z}_{main}, \boldsymbol{m}]), \quad (10)$$

where $\sigma$ denotes the Sigmoid function, and MLPs are two-layer feedforward networks. The refined representation is obtained via $\tilde{\boldsymbol{z}}_{main} = \boldsymbol{z}_{main} + \boldsymbol{g} \odot \boldsymbol{\delta}$, allowing the model to dynamically adjust the semantic features based on the missingness configuration.

**Path 2: Cross-Modal Pattern Recovery.** We trace back the hidden states corresponding to each modality from $\mathbf{H}_{out}$ and perform temporal pooling to obtain modal vectors $\boldsymbol{h}_v, \boldsymbol{h}_a, \boldsymbol{h}_f$. We stack these as $\mathbf{H}_{stack}$ and apply self-attention to capture implicit correlations:

$$\mathbf{Z}_{attn} = \text{Attention}(\mathbf{H}_{stack}, \mathbf{H}_{stack}, \mathbf{H}_{stack}). \quad (11)$$

To calibrate feature reliability, we design an adaptive gating network $\mathcal{G}$ and a learnable parameter $\lambda_m$. We define the inference confidence as $\gamma_m = \text{Sigmoid}(\lambda_m)$, where the mixing weight $\alpha_{m_j}$ depends on the modality $m_j$'s availability:

$$\alpha_{m_j} = \boldsymbol{m}_j \cdot 1 + (1 - \boldsymbol{m}_j) \cdot \gamma_{m_j}. \quad (12)$$

If a modality is present ($\boldsymbol{m}_j = 1$), we trust it fully; if missing ($\boldsymbol{m}_j = 0$), we trust the inferred features according to $\gamma_m$. The auxiliary feature vector is aggregated as:

$$\boldsymbol{z}_{aux} = \sum_m \alpha_m \cdot (\boldsymbol{z}^m_{attn} \odot \mathcal{G}(\mathbf{H}_{stack})^m), \quad (13)$$

where $\boldsymbol{z}^m_{attn}$ denotes the row in $\mathbf{Z}_{attn}$ for modality $m$.

**Collaborative Semantics-Dynamics Decoding.** Rather than treating the two paths as independent predictors, we formulate the final prediction as a collaborative decoding process between the *Generative Semantic Space* (from LLM

reasoning) and the *Discriminative Feature Space* (from cross-modal patterns). We introduce a learnable global synergy coefficient $\lambda_{syn}$ to arbitrate the contribution of prior-driven reasoning versus data-driven evidence:

$$\hat{y} = \underbrace{\lambda_{syn} \cdot \mathcal{H}_{sem}(\tilde{z}_{main})}_{\text{Semantic Reasoning}} + \underbrace{(1 - \lambda_{syn}) \cdot \mathcal{H}_{dyn}(z_{aux})}_{\text{Pattern Recovery}}, \quad (14)$$

where $\mathcal{H}_{sem}$ and $\mathcal{H}_{dyn}$ are regression heads projecting high-level semantics and low-level dynamics into the score space. This design effectively balances the reliance on LLMs' inferred context against the robustness of statistical patterns.

### 3.7. Optimization and Regularization

With ground-truth scores $y$, our LIMSSR framework is trained by minimizing the following objective function:

$$\min_{\theta} \ \mathcal{J}(\theta) = \mathbb{E}_{(\mathcal{X}_{obs}, \boldsymbol{m}, y) \sim \mathcal{D}_{train}} \left[ \mathcal{L}(\hat{y}, y) \right], \quad (15)$$

where $\theta$ denotes all trainable parameters, and $\mathcal{L}$ is a composite loss function combining three components:

$$\mathcal{L} = \lambda_{task}\mathcal{L}_{task} + \lambda_{con}\mathcal{L}_{con} + \lambda_{reg}\mathcal{L}_{reg}. \quad (16)$$

Here, $\lambda_{task}$, $\lambda_{con}$, and $\lambda_{reg}$ are hyperparameters balancing the contributions of each loss term. $\mathcal{L}_{task}$ is the standard Mean Squared Error (MSE) loss for the regression task.

**Dual-Path Consistency ($\mathcal{L}_{con}$).** To enforce self-consistency between high-level reasoning and low-level feature aggregation, we minimize the discrepancy between the predictions of the two paths: $\mathcal{L}_{con} = \|\hat{y}_{main} - \hat{y}_{aux}\|_2^2$.

**Token-level Metric Regularization ($\mathcal{L}_{reg}$).** To prevent feature collapse among the $K$ fusion tokens, we apply a token-level diversity-enforcing regularization. We encourage each token $\boldsymbol{h}_i$ to be sufficiently distinct from others by enforcing a margin between its closest and farthest neighbors:

$$\mathcal{L}_{reg} = \sum_{i=1}^{K} [\max_{j \neq i} \text{sim}(\boldsymbol{h}_i, \boldsymbol{h}_j) - \min_{j \neq i} \text{sim}(\boldsymbol{h}_i, \boldsymbol{h}_j) + \delta]_+, \quad (17)$$

where $\text{sim}(\cdot, \cdot)$ denotes cosine similarity, $\delta$ is a margin hyperparameter, which is set to 1. $[\cdot]_+$ means $\max(0, \cdot)$.

## 4. Experiments

### 4.1. Experiment Settings

**Datasets and Metrics.** We evaluate our LIMSSR on three widely-used long-term AQA benchmarks: FS1000 (Xia et al., 2023), Fis-V (Xu et al., 2019), and Rhythmic Gymnastics (RG) (Zeng et al., 2020). Following prior works (Du et al., 2024; Xu et al., 2026), we report Spearman's Rank Correlation ($\rho$) and Mean Squared Error (MSE). $\rho$ measures the rank-order correlation between predicted and ground-truth sequences, while MSE assesses the numerical error. Higher $\rho$ and lower MSE values indicate better performance.

Fisher's z-value is used to calculate the average $\rho$ across different types. More details are in the Appendix A and B.

**Implementation Details.** LIMSSR is compatible with any pre-trained LLMs. We adopt Qwen3-0.6B (Team, 2025) as the default backbone, selecting a sub-1B parameter model to enable edge deployability while retaining scalability for larger models. To preserve the knowledge of LLMs, we employ LoRA (Hu et al., 2022) for fine-tuning. Since new tokens (e.g., `<visual_start>`) are introduced, we set the token embeddings and the LLM head trainable. Experiments are conducted on a single RTX 3090 GPU with PyTorch 2.4.1. Following standard protocols (Wang et al., 2023; Xu et al., 2024d; 2026), we evaluate on the full set $\{v, a, f\}$ and six subsets. Loss weights $\lambda_{task}$, $\lambda_{con}$, and $\lambda_{reg}$ in Equation (16) are set to 10, 1, and 1, with $K = 4$ fusion tokens. For FS1000/Fis-V/RG, we randomly sample 95/124/68 video segments, with a dropout of 0.15/0.15/0.35 to avoid overfitting. We use AdamW with a cosine annealing scheduler (decay rate 0.1), initial learning rate $2 \times 10^{-4}$, and batch size 8. Training epochs follow dataset-specific settings consistent with prior works (Zhou et al., 2024; Xu et al., 2022a; 2026). More details are in the Appendix B.

### 4.2. Comparison with State-of-the-art

**Incomplete Multimodal Scenarios.** We compare our LIMSSR framework with several state-of-the-art (SOTA) methods designed for Incomplete Multimodal Emotion Recognition (IMER), Action Recognition (IMAR), and Incomplete/complete Multimodal AQA (IMAQA/MAQA). For the widely studied IMER, we evaluate GCNet (Lian et al., 2023), IMDer (Wang et al., 2023), MoMKE (Xu et al., 2024d), and SDR-GNN (Fu et al., 2025). For IMAR, we consider ActionMAE (Woo et al., 2023). For IMAQA/MAQA, we include MLP-Mixer (Xia et al., 2023), PAMFN (Zeng & Zheng, 2024), and MCMoE (Xu et al., 2026).

As shown in Table 1, we benchmark LIMSSR against baselines across all six incomplete modality combinations, their average performance, and the full-modality setting on three datasets. LIMSSR consistently achieves superior or highly competitive results across all metrics and missingness patterns. Specifically, it surpasses the current SOTA methods by improving the average Sp. Corr. ($\rho$)/MSE by 0.9%/8.4%, 1.2%/11.3%, and 1.9%/7.6% on the FS1000, Fis-V, and RG datasets, respectively. The distinct improvement in MSE highlights LIMSSR's ability to maintain precise scoring predictions despite missing modality information. Crucially, unlike existing approaches that rely on the unrealistic "Full-modal training" assumption for supervision or distillation, LIMSSR outperforms these supervision-dependent methods even under the more challenging *Training-Time Incomplete Observations*. This confirms the superior adaptability and robustness of our framework for real-world scenarios with

*Table 1.* Comparisons of performance on three benchmarks with incomplete modalities. $v$, $f$, and $a$ refer to the RGB, flow, and audio modalities. "Average" denotes the average result of all six incomplete multimodal combinations. The **bold** / underline indicate the best / second-best results. ♡, ♣, and ♠ mean the evaluated method sources for incomplete/complete multimodal AQA, incomplete multimodal action recognition, and incomplete multimodal emotion recognition. **T-Miss**: Training-time missing modality. $\Delta_{SOTA}$ means the performance increase or decrease of our LIMSSR compared to the best competing methods.

| Datasets | Methods | T-Miss | Testing Condition (Spearman Correlation (↑) / Mean Squared Error (↓)) | | | | | | | |
| --- | --- | --- | --- | --- | --- | --- | --- | --- | --- | --- |
| | | | {v,f} | {v,a} | {f,a} | {v} | {f} | {a} | **Average** | {v,f,a} |
| FS1000 (7-class) | ♣ActionMAE (AAAI'23) | ✗ | 0.775 / 24.66 | 0.766 / 64.13 | 0.556 / 26.51 | 0.761 / 50.64 | 0.462 / 21.47 | 0.458 / 41.66 | 0.651 / 38.18 | 0.809 / 17.96 |
| | ♠GCNet (TPAMI'23) | ✗ | 0.730 / 25.56 | 0.740 / 23.86 | 0.507 / 24.97 | 0.696 / 26.67 | 0.447 / 31.27 | 0.442 / 39.40 | 0.610 / 28.62 | 0.764 / 21.82 |
| | ♠IMDer (NeurIPS'23) | ✗ | 0.760 / 22.34 | 0.745 / 28.46 | 0.573 / 24.86 | 0.724 / 35.99 | 0.424 / 22.92 | 0.488 / 32.56 | 0.636 / 27.86 | 0.788 / 25.95 |
| | ♡MLP-Mixer (AAAI'23) | ✗ | 0.722 / 25.56 | 0.542 / 60.57 | 0.474 / 87.20 | 0.623 / 101.35 | 0.472 / 87.59 | 0.177 / 68.43 | 0.520 / 71.78 | 0.819 / 14.56 |
| | ♡PAMFN (TIP'24) | ✗ | 0.727 / 59.96 | 0.644 / 62.80 | 0.561 / 56.36 | 0.713 / 92.10 | 0.486 / 117.63 | 0.145 / 62.13 | 0.571 / 75.16 | 0.855 / 13.02 |
| | ♠MoMKE (ACMMM'24) | ✗ | 0.798 / 18.86 | 0.805 / 23.88 | 0.541 / 24.96 | 0.785 / 37.96 | 0.398 / 23.31 | 0.499 / 27.53 | 0.668 / 26.08 | 0.819 / 16.85 |
| | ♠SDR-GNN (KBS'25) | ✗ | 0.789 / 17.50 | 0.785 / 25.08 | 0.564 / 22.29 | 0.749 / 28.47 | 0.504 / 29.96 | 0.477 / 25.46 | 0.665 / 24.79 | 0.817 / 15.91 |
| | ♡MCMoE (AAAI'26) | ✗ | 0.845 / 12.66 | 0.882 / 11.85 | **0.738** / **14.88** | 0.845 / 13.64 | **0.650** / 22.47 | 0.615 / 16.72 | 0.782 / 15.37 | 0.881 / 11.53 |
| | **LIMSSR** (This Paper) | ✓ | **0.854** / **12.51** | **0.891** / **10.54** | 0.709 / 14.98 | **0.853** / **12.50** | 0.618 / **18.45** | **0.687** / **15.51** | **0.789** / **14.08** | **0.891** / **10.44** |
| | $\Delta_{SOTA}$ | - | ↑1.1% / ↓1.2% | ↑1.0% / ↓11.1% | ↓3.9% / ↑0.7% | ↑0.9% / ↓8.4% | ↓4.9% / ↓17.9% | ↑11.7% / ↓7.2% | ↑0.9% / ↓8.4% | ↑1.1% / ↓9.5% |
| Fis-V (2-class) | ♣ActionMAE (AAAI'23) | ✗ | 0.704 / 33.54 | 0.678 / 27.61 | 0.575 / 25.55 | 0.616 / 40.07 | 0.484 / 24.87 | 0.486 / 29.29 | 0.597 / 30.16 | 0.698 / 17.34 |
| | ♠GCNet (TPAMI'23) | ✗ | 0.738 / 19.86 | 0.656 / 21.32 | 0.594 / 21.72 | 0.667 / 20.87 | 0.602 / 19.61 | 0.455 / 34.77 | 0.626 / 23.03 | 0.698 / 16.93 |
| | ♠IMDer (NeurIPS'23) | ✗ | 0.748 / 15.19 | 0.658 / 22.46 | 0.568 / 23.99 | 0.675 / 25.45 | 0.618 / 26.38 | 0.405 / 31.96 | 0.622 / 24.27 | 0.703 / 17.02 |
| | ♡MLP-Mixer (AAAI'23) | ✗ | 0.732 / 30.34 | 0.651 / 46.70 | 0.572 / 26.96 | 0.618 / 48.46 | 0.546 / 27.18 | 0.325 / 67.25 | 0.586 / 41.15 | 0.772 / 13.97 |
| | ♡PAMFN (TIP'24) | ✗ | 0.801 / 33.49 | 0.661 / 54.34 | 0.622 / 110.50 | 0.644 / 84.93 | 0.616 / 110.42 | 0.141 / 86.16 | 0.610 / 79.97 | 0.822 / 15.33 |
| | ♠MoMKE (ACMMM'24) | ✗ | 0.754 / 14.84 | 0.689 / 20.60 | 0.646 / 19.62 | 0.684 / 23.16 | 0.654 / 22.46 | 0.497 / 29.09 | 0.660 / 21.63 | 0.747 / 17.30 |
| | ♠SDR-GNN (KBS'25) | ✗ | 0.752 / 14.99 | 0.680 / 20.62 | 0.619 / 20.89 | 0.689 / 20.26 | 0.648 / 21.50 | 0.479 / 32.13 | 0.651 / 21.73 | 0.733 / 16.45 |
| | ♡MCMoE (AAAI'26) | ✗ | 0.813 / 11.02 | 0.787 / 14.64 | **0.727** / 17.41 | 0.765 / 15.14 | **0.698** / **15.39** | 0.557 / 28.54 | 0.734 / 17.02 | 0.829 / 12.15 |
| | **LIMSSR** (This Paper) | ✓ | **0.824** / **10.17** | **0.822** / **11.31** | 0.693 / **16.16** | **0.790** / **11.91** | 0.677 / 17.19 | **0.578** / 23.80 | **0.743** / **15.09** | **0.841** / **10.19** |
| | $\Delta_{SOTA}$ | - | ↑1.4% / ↓7.7% | ↑4.4% / ↓22.7% | ↓4.7% / ↓7.2% | ↑3.3% / ↓21.3% | ↓3.0% / ↑11.7% | ↑3.8% / ↓16.6% | ↑1.2% / ↓11.3% | ↑1.4% / ↓16.1% |
| RG (4-class) | ♣ActionMAE (AAAI'23) | ✗ | 0.724 / 7.30 | 0.621 / 8.76 | 0.545 / 10.39 | 0.689 / 12.41 | 0.521 / 11.29 | 0.251 / 16.84 | 0.575 / 11.16 | 0.709 / 7.01 |
| | ♠GCNet (TPAMI'23) | ✗ | 0.738 / 6.69 | 0.638 / 7.95 | 0.556 / 11.75 | 0.701 / 8.12 | 0.568 / 36.01 | 0.225 / 15.20 | 0.591 / 14.29 | 0.716 / 6.45 |
| | ♠IMDer (NeurIPS'23) | ✗ | 0.746 / 6.03 | 0.646 / 7.55 | 0.569 / 8.80 | 0.699 / 7.59 | 0.596 / 9.35 | 0.206 / 14.19 | 0.598 / 8.92 | 0.724 / 6.37 |
| | ♡MLP-Mixer (AAAI'23) | ✗ | 0.733 / 7.23 | 0.614 / 14.09 | 0.485 / 10.18 | 0.655 / 9.67 | 0.566 / 11.45 | 0.244 / 16.01 | 0.567 / 11.44 | 0.754 / 7.48 |
| | ♡PAMFN (TIP'24) | ✗ | 0.764 / 6.78 | 0.616 / 38.87 | 0.448 / 122.11 | 0.658 / 39.86 | 0.483 / 123.44 | 0.131 / 151.69 | 0.543 / 80.46 | 0.819 / 6.64 |
| | ♠MoMKE (ACMMM'24) | ✗ | 0.762 / 5.69 | 0.656 / 7.97 | 0.629 / 9.39 | 0.693 / 8.42 | 0.621 / 10.08 | 0.264 / 13.25 | 0.623 / 9.13 | 0.747 / 6.18 |
| | ♠SDR-GNN (KBS'25) | ✗ | 0.758 / 6.08 | 0.655 / 7.38 | 0.612 / 9.40 | 0.727 / 7.77 | 0.591 / 9.80 | 0.264 / 13.53 | 0.621 / 8.99 | 0.742 / 6.35 |
| | ♡MCMoE (AAAI'26) | ✗ | 0.822 / 5.33 | 0.781 / 5.83 | **0.699** / 8.15 | 0.767 / 6.25 | 0.662 / 8.59 | 0.278 / 13.20 | 0.697 / 7.89 | 0.842 / **4.85** |
| | **LIMSSR** (This Paper) | ✓ | **0.825** / **4.88** | **0.809** / **5.49** | 0.656 / **7.72** | **0.781** / **5.61** | **0.671** / **7.17** | **0.359** / **12.86** | **0.710** / **7.29** | **0.855** / 5.07 |
| | $\Delta_{SOTA}$ | - | ↑0.4% / ↓8.4% | ↑3.6% / ↓5.8% | ↓6.2% / ↓5.3% | ↑1.8% / ↓10.2% | ↑1.4% / ↓16.5% | ↑29.1% / ↓2.6% | ↑1.9% / ↓7.6% | ↑1.5% / ↑4.5% |

inherently imperfect data. We attribute this success to our reformulation of IMAQA as a conditional sequence-to-score reasoning task, which effectively exploits the semantic reasoning and cross-modal understanding of LLMs to infer valuable representations for missing data.

**Complete Modality Scenarios.** We further benchmark LIMSSR against SOTA unimodal and multimodal AQA methods under the standard complete modality setting. As shown in Table 2, LIMSSR consistently achieves superior or highly competitive performance across all benchmarks. Notably, LIMSSR represents a pioneering attempt in AQA to address *Training-Time Incomplete Observations*. Despite lacking the full-modal supervision available to multimodal methods (e.g., MLP-Mixer (Xia et al., 2023), SGN (Du et al., 2024), PAMFN (Zeng & Zheng, 2024), VATP-Net (Gedamu et al., 2025), and MCMoE (Xu et al., 2026)), LIMSSR sets a new state-of-the-art on FS1000 and Fis-V, improving average metrics by 1.1%/9.5% and 1.4%/16.1%. On the RG dataset, it achieves the best Sp. Corr. ($\rho$) and the second-best MSE, yielding an average improvement of 1.3%/7.0% across the three datasets. These results indicate that LIMSSR effectively leverages partial observations to infer complementary latent semantics and model precise cross-modal patterns, even when trained with incomplete data. Furthermore, compared to leading unimodal methods

(T²CR (Ke et al., 2024), CoFInAl (Zhou et al., 2024), and ASGTN (Liu et al., 2025a)), our method exhibits significant performance margins across all benchmarks, confirming its ability to mitigate the adverse effects of missingness and successfully harness the strengths of multimodal learning.

### 4.3. Ablation Study

**Effect of Components.** LIMSSR comprises three core modules: Prompt-Guided Context-Aware Modality Imputation (PCMI), LLM-Driven Multidimensional Representation Fusion (LMRF), and Mask-Aware Dual-Path Aggregation (MDA). To validate their individual contributions, we conduct ablation studies on the FS1000 and Fis-V datasets, as summarized in Table 3. We establish a *Baseline* that directly applies cross-modal attention to available modalities, followed by an MLP regressor. The top half of the table illustrates the progressive addition of components, where "+LLM" denotes processing available features without prompts via the LLM backbone before prediction. The bottom half isolates the impact of removing specific modules. As observed, simply integrating the LLM yields substantial gains, verifying the benefit of its reasoning priors. Incorporating PCMI significantly boosts performance by structuring missingness as a fill-in-the-blank task rather than zero-padding. LMRF further refines feature integration into

*Table 2.* Comparisons of performance on three benchmarks with complete modalities. The **bold** / underline indicate the best / second-best results. † means the dataset does not include this category. ◇ and ♡ mean unimodal and multimodal AQA methods. **T-Miss**: Training-time missing modality. $\Delta_{SOTA}$ means the performance increase or decrease of our LIMSSR compared to the best competing methods.

| Datasets | Methods | T-Miss | Assessment Category (Spearman Correlation (↑) / Mean Squared Error (↓)) | | | | | | | |
|---|---|---|---|---|---|---|---|---|---|---|
| | | | TES | PCS | SS | TR | PE | CO | IN | Average |
| FS1000 | ♡MLP-Mixer (AAAI'23) | ✗ | 0.880 / 81.24 | 0.820 / 9.47 | 0.800 / 0.35 | 0.810 / 0.35 | 0.800 / 0.62 | 0.810 / 0.37 | 0.810 / 0.39 | 0.821 / 13.26 |
| | ◇T²CR (Inf. Sci.'24) | ✗ | 0.863 / 107.59 | 0.794 / 15.26 | 0.833 / 0.61 | 0.837 / 0.48 | 0.823 / 0.69 | 0.843 / 0.57 | 0.804 / 0.42 | 0.829 / 17.95 |
| | ◇CoFInAl (IJCAI'24) | ✗ | 0.835 / 81.65 | 0.830 / 16.05 | 0.838 / 0.56 | 0.836 / 0.63 | 0.814 / 0.71 | 0.829 / 0.41 | 0.819 / 0.54 | 0.829 / 14.36 |
| | ♡SGN (TMM'24) | ✗ | 0.890 / 79.08 | 0.850 / 8.40 | 0.840 / 0.31 | 0.850 / 0.32 | 0.820 / 0.61 | 0.850 / 0.33 | 0.830 / 0.37 | 0.849 / 12.77 |
| | ♡PAMFN (TIP'24) | ✗ | 0.874 / 78.42 | 0.854 / 9.72 | 0.848 / 0.54 | 0.866 / 0.58 | 0.857 / 0.69 | 0.834 / 0.53 | 0.846 / 0.64 | 0.855 / 13.02 |
| | ◇ASGTN (TCSVT'25) | ✗ | 0.873 / 94.83 | 0.849 / 14.58 | 0.858 / 0.35 | 0.856 / 0.38 | 0.832 / 0.68 | 0.846 / 0.39 | 0.835 / 0.47 | 0.850 / 15.95 |
| | ♡MCMoE (AAAI'26) | ✗ | 0.900 / 71.69 | 0.872 / 7.43 | 0.876 / **0.26** | 0.882 / **0.26** | 0.872 / **0.52** | 0.886 / **0.26** | 0.874 / 0.29 | 0.881 / 11.53 |
| | **LIMSSR** (This Paper) | ✓ | **0.907 / 64.58** | **0.892 / 6.84** | **0.894 / 0.26** | **0.887** / 0.27 | **0.885 / 0.52** | **0.889** / 0.28 | **0.877 / 0.28** | **0.891 / 10.44** |
| | $\Delta_{SOTA}$ | - | ↑0.8% / ↓9.9% | ↑2.3% / ↓7.9% | ↑2.1% / ↓0.0% | ↑0.6% / ↑3.8% | ↑1.5% / ↓0.0% | ↑0.3% / ↑7.7% | ↑0.3% / ↓3.4% | ↑1.1% / ↓9.5% |
| Fis-V | ♡MLP-Mixer (AAAI'23) | ✗ | 0.680 / 19.57 | 0.820 / 7.96 | † | † | † | † | † | 0.759 / 13.77 |
| | ◇T²CR (Inf. Sci.'24) | ✗ | 0.702 / 20.84 | 0.811 / 8.10 | † | † | † | † | † | 0.762 / 14.47 |
| | ◇CoFInAl (IJCAI'24) | ✗ | 0.716 / 20.76 | 0.843 / 7.91 | † | † | † | † | † | 0.788 / 14.34 |
| | ♡SGN (TMM'24) | ✗ | 0.700 / 19.05 | 0.830 / 7.96 | † | † | † | † | † | 0.773 / 13.51 |
| | ♡PAMFN (TIP'24) | ✗ | 0.754 / 22.50 | 0.872 / 8.16 | † | † | † | † | † | 0.822 / 15.33 |
| | ♡VATP-Net (TCSVT'25) | ✗ | 0.702 / - | 0.863 / - | † | † | † | † | † | 0.796 / - |
| | ◇ASGTN (TCSVT'25) | ✗ | 0.703 / 21.37 | 0.845 / 10.75 | † | † | † | † | † | 0.784 / 16.06 |
| | ♡MCMoE (AAAI'26) | ✗ | 0.759 / 18.50 | **0.880** / 5.81 | † | † | † | † | † | 0.829 / 12.15 |
| | **LIMSSR** (This Paper) | ✓ | **0.792 / 14.82** | **0.880 / 5.55** | † | † | † | † | † | **0.841 / 10.19** |
| | $\Delta_{SOTA}$ | - | ↑4.3% / ↓19.9% | ↑0.0% / ↓4.5% | † | † | † | † | † | ↑1.4% / ↓16.1% |
| | | | Ball | Clubs | Hoop | Ribbon | † | † | † | Average |
| RG | ♡MLP-Mixer (AAAI'23) | ✗ | 0.677 / 6.75 | 0.708 / 5.81 | 0.778 / 5.94 | 0.706 / 6.87 | † | † | † | 0.719 / 6.34 |
| | ◇T²CR (Inf. Sci.'24) | ✗ | 0.750 / 7.18 | 0.800 / 5.63 | 0.794 / 5.88 | 0.827 / 5.91 | † | † | † | 0.794 / 6.13 |
| | ◇CoFInAl (IJCAI'24) | ✗ | 0.809 / **5.07** | 0.806 / 5.19 | 0.804 / 6.37 | 0.810 / 6.30 | † | † | † | 0.807 / 5.73 |
| | ♡PAMFN (TIP'24) | ✗ | 0.757 / 6.24 | 0.825 / 7.45 | 0.836 / **5.21** | 0.846 / 7.67 | † | † | † | 0.819 / 6.64 |
| | ♡VATP-Net (TCSVT'25) | ✗ | 0.800 / - | 0.810 / - | 0.780 / - | 0.769 / - | † | † | † | 0.790 / - |
| | ◇ASGTN (TCSVT'25) | ✗ | 0.792 / 6.60 | 0.825 / 5.66 | 0.784 / **5.21** | 0.793 / 6.75 | † | † | † | 0.799 / 6.06 |
| | ♡MCMoE (AAAI'26) | ✗ | 0.806 / 5.66 | 0.815 / **4.22** | 0.845 / 5.62 | 0.890 / **3.89** | † | † | † | 0.842 / **4.85** |
| | **LIMSSR** (This Paper) | ✓ | **0.813** / 5.50 | **0.830** / 4.69 | **0.850** / 5.33 | **0.908** / 4.76 | † | † | † | **0.855** / 5.07 |
| | $\Delta_{SOTA}$ | - | ↑0.5% / ↑8.5% | ↑0.6% / ↑18.0% | ↑0.6% / ↑2.3% | ↑2.0% / ↑22.4% | † | † | † | ↑1.5% / ↑4.5% |

*Table 3.* Ablation results on the FS1000 and Fis-V. The top half adds our components in order, and the bottom half individually removes one. Results are shown by $\rho$(↑) / MSE(↓).

| Settings | FS1000 | | Fis-V | |
|---|---|---|---|---|
| | Average | $\{v,f,a\}$ | Average | $\{v,f,a\}$ |
| Baseline | 0.528 / 37.81 | 0.755 / 29.07 | 0.558 / 59.79 | 0.718 / 18.67 |
| + LLM | 0.601 / 29.90 | 0.791 / 24.38 | 0.648 / 37.33 | 0.744 / 15.18 |
| + PCMI | 0.687 / 24.02 | 0.838 / 20.11 | 0.689 / 24.62 | 0.786 / 13.07 |
| + LMRF (w/o MDA) | 0.768 / 18.18 | 0.866 / 15.42 | 0.720 / 17.14 | 0.811 / 12.04 |
| **+ MDA (Ours)** | **0.789 / 14.08** | **0.891 / 10.44** | **0.743 / 15.09** | **0.841 / 10.19** |
| w/o LMRF | 0.732 / 20.39 | 0.837 / 16.42 | 0.685 / 20.85 | 0.793 / 14.62 |
| w/o PCMI | 0.751 / 19.43 | 0.844 / 17.94 | 0.699 / 18.87 | 0.803 / 13.48 |
| w/o $\mathcal{L}_{con}$ | 0.759 / 17.82 | 0.862 / 15.50 | 0.711 / 17.98 | 0.814 / 13.13 |
| w/o $\mathcal{L}_{reg}$ | 0.773 / 17.47 | 0.878 / 14.08 | 0.732 / 18.32 | 0.829 / 11.76 |

*Table 4.* Controlled analysis of the contribution of LLM initialization, fine-tuning, and LIMSSR designs on FS1000 and Fis-V. Results are shown by $\rho$(↑) / MSE(↓).

| Settings | FS1000 | | Fis-V | |
|---|---|---|---|---|
| | Average | $\{v,f,a\}$ | Average | $\{v,f,a\}$ |
| Baseline | 0.528 / 37.81 | 0.755 / 29.07 | 0.558 / 59.79 | 0.718 / 18.67 |
| Random LLM | 0.574 / 34.72 | 0.769 / 30.62 | 0.606 / 50.23 | 0.731 / 16.85 |
| Frozen LLM | 0.592 / 30.82 | 0.784 / 27.40 | 0.633 / 43.56 | 0.740 / 16.08 |
| LLM+LoRA | 0.601 / 29.90 | 0.791 / 24.38 | 0.648 / 37.33 | 0.744 / 15.18 |
| LIMSSR w/ Random LLM | 0.698 / 19.86 | 0.847 / 18.28 | 0.680 / 20.99 | 0.788 / 14.81 |
| **LIMSSR (Ours)** | **0.789 / 14.08** | **0.891 / 10.44** | **0.743 / 15.09** | **0.841 / 10.19** |

latent slots, while MDA achieves the best stability by mitigating hallucinations via uncertainty calibration. The consistent degradation upon removing any component confirms that all modules are essential for LIMSSR's robustness.

**Effect of Loss Terms.** We further analyze the auxiliary losses $\mathcal{L}_{con}$ and $\mathcal{L}_{reg}$. As shown in the bottom half of Table 3, removing either loss leads to noticeable performance drops. Excluding $\mathcal{L}_{con}$ disrupts the alignment between high-level reasoning and low-level feature statistics, reducing prediction reliability. Meanwhile, omitting $\mathcal{L}_{reg}$ results in feature collapse among the fusion tokens, hindering the LLM's ability to encode diverse evaluation dimensions. These results confirm that both losses are essential for constraining the latent space and ensuring robust multimodal representation learning.

**Contribution of LLM Priors and LIMSSR Designs.** We conduct a controlled analysis in Table 4 to disentangle the effects of model scale, pre-training, fine-tuning, and our architectural designs. Replacing the baseline with a randomly initialized LLM-sized backbone ("Random LLM") yields noticeable improvements, suggesting increased capacity inherently accommodates severe data missingness. Using pre-trained weights ("Frozen LLM") and LoRA adaptation ("LLM+LoRA") delivers further gains, confirming the value of semantic priors and task-specific adaptation. Strikingly, integrating our framework modules (PCMI, LMRF, and MDA) drives the most substantial performance surge. Furthermore, "LIMSSR w/ Random LLM" significantly outperforms the vanilla "Random LLM," proving our designs provide a robust, discriminative reasoning structure beyond passive imputation. Finally, the gap between "LIMSSR w/ Random LLM" and the full "LIMSSR" confirms that our modules synergize optimally with the sequence-reasoning

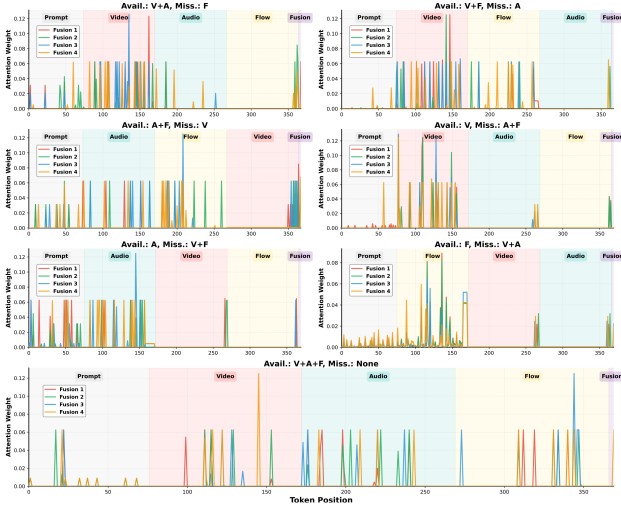

*Figure 3.* Visualization of fusion token attention weights across different modality combinations on the FS1000 (PCS).

capabilities of pre-trained LLMs to address training-time incomplete observations. More ablation studies are detailed in Appendix C.

### 4.4. Visualization Analysis

In Figure 3, we visualize the attention weights of the $K$ fusion tokens in the LLM's final layer across input modalities. We observe that LIMSSR dynamically reallocates attention according to modality availability. When a modality is missing, tokens shift focus to exploit the remaining available context, while under full modalities attention is more evenly distributed. Rather than performing pixel-level reconstruction, our method attends to key semantic regions, such as the aggregated global semantics at the end of imputed sequences, supporting our premise of reasoning-based latent imputation. Moreover, consistent with our regularization strategy $\mathcal{L}_{reg}$, different fusion tokens attend to distinct patterns, confirming their capacity to encode diverse evaluation dimensions. More visualizations are in the Appendix D.

### 5. Conclusion

This paper presents LIMSSR, a novel LLM-driven framework for incomplete multimodal learning under challenging training-time incomplete observations. By reformulating the problem as a conditional sequence reasoning task, LIMSSR effectively leverages the inherent reasoning capabilities of LLMs to infer latent semantics for missing modalities without relying on complete training data. We introduce Prompt-Guided Context-Aware Modality Imputation to structure the reasoning process, and Multidimensional Representation Fusion to integrate multimodal information into task-specific latent slots. To mitigate potential hallucinations, a Mask-Aware Dual-Path Aggregation mechanism dynamically cal-

ibrates the reliance between high-level inference and low-level feature aggregation. Extensive experiments on three public benchmarks demonstrate that LIMSSR achieves state-of-the-art performance, validating the efficacy of leveraging LLMs for data-efficient multimodal learning in real-world scenarios marked by systemic data missingness.

**Future work** considers extending this paradigm to diverse multimodal tasks and incorporating explicit physical constraints to further enhance reasoning reliability. In addition, we will explore validating the generalization of LIMSSR across other incomplete multimodal tasks and introducing physical constraints to enhance reasoning reliability.

### Acknowledgements

This work was supported by the grants from the National Key Research and Development Plan of China (2021YFB3600503), National Natural Science Foundation of China (61972097, U21A20472), Major Scientific Research Project for Technology Promotes Police (2025YZ040003, 2024YZ040001), and Natural Science Foundation of Fujian Province (2025J01536).

### Impact Statement

This paper presents work whose goal is to advance the field of Machine Learning. There are many potential societal consequences of our work, none which we feel must be specifically highlighted here.

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

# A. Scenario Characteristics and Datasets

We instantiate our proposed framework within the context of long-term Action Quality Assessment (AQA). Long-term AQA aims to evaluate skill levels by analyzing an agent's performance over extended durations. This process typically involves extracting spatiotemporal features from video segments via pre-trained backbones to form long-term sequence representations, making it inherently suitable for reformulating incomplete multimodal learning as a conditional sequence reasoning problem.

We select long-term AQA as the experimental testbed for three primary reasons: 1) **Broad Applicability and Multimodality:** Long-term AQA is critical in diverse scenarios, including competitive sports, medical rehabilitation, professional certification, and human-computer interaction. It naturally involves multimodal data (e.g., RGB (Xu et al., 2022b; Zhou et al., 2023; Xu et al., 2025a), audio (Xia et al., 2023; Zeng & Zheng, 2024; Xu et al., 2025a), sensor data (Zeng & Zheng, 2024; Xu et al., 2026), skeletons (Bruce et al., 2024), and text (Zhang et al., 2024; Xu et al., 2025a)), providing a rich environment for investigating multimodal and incomplete multimodal learning. 2) **Real-World Relevance to Missingness:** Data acquisition in AQA often varies by scene and difficulty, frequently suffering from systemic missingness due to sensor failures, occlusion (Xu et al., 2025b), or privacy constraints (Bruce et al., 2024). This aligns precisely with the challenge of learning under training-time incomplete observations addressed in this work. 3) **Complexity and Reasoning Requirements:** Accurate quality assessment requires modeling complex long-term dependencies, abstract high-level semantics, and subtle fine-grained motion differences. These challenges effectively benchmark the capability of LIMSSR to infer and fuse incomplete multimodal information via LLM-driven reasoning. Consequently, long-term AQA serves as an ideal platform to validate the applicability and robustness of LIMSSR in realistic incomplete multimodal learning scenarios.

Available public long-term AQA benchmarks are employed for extensive evaluation: FS1000 (Xia et al., 2023), Fis-V (Xu et al., 2019), and Rhythmic Gymnastics (RG) (Zeng et al., 2020). These datasets encompass distinct sport types (figure skating and rhythmic gymnastics) and diverse evaluation criteria (spanning over a dozen action classes and score categories), providing comprehensive multimodal data including visual, audio, and optical flow streams. Brief descriptions of these datasets follow:

**FS1000.** The FS1000 dataset serves as a large-scale benchmark for figure skating, comprising 1,247 videos in total (1,000 for training and 247 for validation). It spans eight distinct competition categories, including short programs and free skating for men, ladies, and pairs, as well as rhythm and free dances for ice dance. The videos, recorded at 25 fps, possess an average length of roughly 5,000 frames. Annotations are provided for the Total Element Score (TES) and the Total Program Component Score (PCS), with the latter further broken down into five sub-metrics: Skating Skills (SS), Transitions (TR), Performance (PE), Composition (CO), and Interpretation (IN). As a pioneering dataset facilitating audio-visual analysis in this domain, FS1000 allows for multimodal evaluation. Consistent with established protocols (Xia et al., 2023; Du et al., 2024; Xu et al., 2026), we develop separate models for each specific scoring criterion.

**Figure Skating Video (Fis-V).** Fis-V collects 500 performance videos focused exclusively on the ladies' singles short program. Videos average 2.9 minutes in duration with a frame rate of 25 fps. The dataset adopts a standard partitioning scheme, allocating 400 samples for training and 100 for testing. Ground truth labels correspond to the official scoring rules, covering both Total Element Score (TES) and Total Program Component Score (PCS). Following the methodology of prior studies (Xu et al., 2019; 2022a; Xia et al., 2023; Du et al., 2024; Zhou et al., 2024; Zeng & Zheng, 2024; Xu et al., 2026), we train independent regressors for each score category.

**Rhythmic Gymnastics (RG).** The RG dataset comprises 1,000 sequences illustrating rhythmic gymnastics routines across four apparatus disciplines: ball, clubs, hoop, and ribbon. Each clip is captured at 25 fps with a duration of approximately 1.6 minutes. Partitioning is performed with a 4:1 ratio, resulting in 200 training and 50 evaluation samples for each apparatus type. In line with previous literature (Zeng et al., 2020; Xu et al., 2022a; Zhou et al., 2024; Zeng & Zheng, 2024; Xu et al., 2026), we train apparatus-specific models based on the standard scoring annotations provided.

# B. More Experiment Settings

**Evaluation Metrics.** Following prior works (Xia et al., 2023; Du et al., 2024; Xu et al., 2025c;a; 2026), we use Spearman's Rank Correlation ($\rho$) and Mean Squared Error (MSE) as evaluation metrics. $\rho$ measures the rank-order correlation between

predicted sequences $\hat{q}$ and ground-truth $q$, while MSE assesses the accuracy of predicted scores $\hat{y}$:

$$\rho = \frac{\sum_i (q_i - \bar{q}) (\hat{q}_i - \bar{\hat{q}})}{\sqrt{\sum_i (q_i - \bar{q})^2 \sum_i (\hat{q}_i - \bar{\hat{q}})^2}}, \tag{18}$$

$$\text{MSE} = \frac{1}{N} \sum_{i=1}^{N} (y_i - \hat{y}_i)^2. \tag{19}$$

Higher $\rho$ values and lower MSE values indicate better performance. Fisher's z-value is used to calculate the average Spearman correlation across different categories. We compute average MSE by arithmetic averaging across settings.

**Feature Extraction Details.** As detailed in Section 3.3, consistent with prior works (Xia et al., 2023; Du et al., 2024; Zeng & Zheng, 2024; Xu et al., 2026), we extract visual, optical flow, and audio features using pre-trained backbones. To ensure fair comparison with existing baselines, we utilize the pre-extracted features provided by these works. Specifically, for the FS1000 dataset, we adopt visual and audio features from MLP-Mixer (Xia et al., 2023) and optical flow features following MCMoE (Xu et al., 2026). For the Fis-V and RG datasets, we employ the pre-extracted visual, audio, and optical flow features provided by PAMFN (Zeng & Zheng, 2024). By adhering to these settings, we maintain input representations that are strictly consistent with the state-of-the-art baseline MCMoE (Xu et al., 2026). All features have been optimized on their respective pre-training tasks, ensuring high-quality input representations.

**Training Strategy and Epoch Configurations.** We employ a cosine annealing scheduler to adaptively modulate the learning rate throughout the training process. To ensure optimal convergence, training epochs are tailored to specific datasets and metrics, consistent with prior studies (Zeng et al., 2020; Xu et al., 2022a; Zeng & Zheng, 2024; Zhou et al., 2024; Xu et al., 2026). Specifically, on the FS1000 dataset, the epoch counts are set as follows: TES (160), PCS (100), SS (130), TR (90), PE (130), CO (200), and IN (290). For the Fis-V benchmark, we train for 270 epochs on TES and 140 epochs on PCS. Regarding the RG dataset, the training durations are 80 epochs for Ball, 100 for Clubs, 100 for Hoop, and 270 for Ribbon. These configurations are chosen to align with the distinct data distributions of each category.

**Score Normalization Protocol.** Given the varied score ranges across different datasets provided by the organizers, regression robustness can be compromised. Following standard protocols (Zeng et al., 2020; Xu et al., 2022a; Zeng & Zheng, 2024; Zhou et al., 2024; Xu et al., 2026), we normalize the ground-truth scores into a unified interval $[0, 1]$ via a scaling factor $\xi$. Formally, a raw score $y_i$ is transformed into a normalized label $\hat{y}_i = y_i/\xi$, where $\xi$ is derived from the maximum possible score in the training partition. In our implementation, $\xi$ is set to 130 for FS1000-TES, 60 for FS1000-PCS, and 10 for the remaining FS1000 sub-metrics (SS, TR, PE, CO, IN). For Fis-V, $\xi$ values are 45 (TES) and 40 (PCS), while RG uses $\xi = 25$. During evaluation, predicted scores are rescaled by multiplying by $\xi$ to calculate the MSE metric, ensuring fair comparison with baselines.

**Computational Environment.** All experiments are conducted on a workstation equipped with an NVIDIA RTX 3090 GPU and a 2.40GHz CPU, running PyTorch 2.4.1 and CUDA 12.4. As an efficiency reference, training on FS1000 for 100 epochs with a batch size of 8 utilizing pre-extracted visual, flow, and audio features requires approximately four hours.

**Architectural Specifications.** In LIMSSR, the modality-specific projection blocks consist of two $1{\times}1$ convolutional layers, followed by Batch Normalization, ReLU activation, and Dropout. The residual corrector $\text{MLP}_{res}$ comprises two fully connected (FC) layers with GELU non-linearity. The gating network $\text{MLP}_{gate}$ similarly uses two FC layers but concludes with a Sigmoid activation. The cross-modal attention mechanism is implemented as a single-layer four-head self-attention module. For adaptive modal weighting, the gate module consists of two FC layers, GELU activation, and a final Softmax function. Both regression heads, $\mathcal{H}_{sem}$ and $\mathcal{H}_{dyn}$, are composed of two FC layers, Layer Normalization, GELU activation, and Dropout. The learnable parameters $\lambda_{syn}$ and $\lambda_m$ are initialized to 0.7 and 0.6, respectively. For LoRA fine-tuning, we set the rank to 16, alpha to 32, and dropout to 0.1.

## C. More Ablation Studies

In this section, we will further conduct some ablation studies to determine the experimental details. Unless otherwise stated, all ablation studies are performed on the FS1000 and Fis-V datasets.

**Effect of LLM Backbones.** We emphasize that LIMSSR does not rely on any complete-modality teacher, reconstruction targets, or external retrieval resources; it only leverages the priors already contained in an off-the-shelf pretrained Large

*Table 5.* Ablation study on different LLM backbones on the FS1000 and Fis-V datasets.

| Methods | FS1000 | | Fis-V | |
|---------|--------|--------|-------|--------|
| | Average | $\{v, f, a\}$ | Average | $\{v, f, a\}$ |
| LIMSSR-Qwen2.5-0.5B | 0.771 / 15.65 | 0.884 / 12.01 | 0.728 / 16.54 | 0.832 / 11.56 |
| LIMSSR-MiniCPM-1B | 0.782 / 14.36 | 0.888 / 11.02 | 0.739 / 15.59 | **0.844** / 10.25 |
| LIMSSR-Llama3.2-1B | 0.775 / 16.21 | 0.878 / 12.54 | 0.742 / 17.50 | 0.832 / 12.37 |
| LIMSSR-Gemma3-1B | 0.768 / 15.30 | 0.879 / 12.82 | 0.725 / 17.29 | 0.827 / 13.50 |
| **LIMSSR-Qwen3-0.6B** | **0.789 / 14.08** | **0.891 / 10.44** | **0.743 / 15.09** | 0.841 / **10.19** |

*Table 6.* Ablation study on the number of fusion tokens $K$ in LIMSSR on the FS1000 and Fis-V datasets.

| #$K$ | FS1000 | | Fis-V | |
|------|--------|--------|-------|--------|
| | Average | $\{v, f, a\}$ | Average | $\{v, f, a\}$ |
| 2 | 0.762 / 19.45 | 0.872 / 14.32 | 0.721 / 18.76 | 0.820 / 12.98 |
| 3 | 0.775 / 17.89 | 0.880 / 13.21 | 0.732 / 17.45 | 0.829 / 12.34 |
| 4 | **0.789** / 14.08 | **0.891 / 10.44** | **0.743 / 15.09** | 0.841 / **10.19** |
| 5 | 0.783 / **14.00** | 0.888 / 11.12 | 0.738 / 16.22 | **0.843** / 11.05 |
| 6 | 0.779 / 15.67 | 0.885 / 12.01 | 0.734 / 17.01 | 0.830 / 11.67 |

*Table 7.* Ablation analysis of the Dual-Path modules within the MDA mechanism on the FS1000 and Fis-V datasets.

| Settings | FS1000 | | Fis-V | |
|----------|--------|--------|-------|--------|
| | Average | $\{v, f, a\}$ | Average | $\{v, f, a\}$ |
| Simple Fusion (w/o MDA) | 0.768 / 18.18 | 0.866 / 15.42 | 0.720 / 17.14 | 0.811 / 12.04 |
| Path 1 Only | 0.778 / 14.95 | 0.885 / 11.23 | 0.735 / 16.02 | 0.835 / 10.85 |
| Path 2 Only | 0.769 / 15.80 | 0.878 / 12.05 | 0.724 / 16.65 | 0.824 / 11.60 |
| **Full MDA** | **0.789 / 14.08** | **0.891 / 10.44** | **0.743 / 15.09** | **0.841 / 10.19** |

Language Model (LLM) through prompt-conditioned sequence modeling. Here, we investigate the impact of different LLM backbones on the performance of LIMSSR. Specifically, in addition to our default *Qwen3-0.6B* (Team, 2025), we evaluate *Qwen2.5-0.5B* (Qwen et al., 2025), *MiniCPM-1B* (Hu et al., 2024), *Llama3.2-1B* (Grattafiori et al., 2024), and *Gemma3-1B* (Team et al., 2025). As presented in Table 5, *Qwen3-0.6B* consistently achieves superior results across both benchmarks, confirming its robust reasoning capabilities for incomplete multimodal learning. While *MiniCPM-1B* yields competitive performance, *Llama3.2-1B* and *Gemma3-1B* exhibit comparatively modest results. These disparities likely stem from variations in pre-training corpora and architectural designs. Consequently, we adopt *Qwen3-0.6B* as the default backbone for LIMSSR to maximize reasoning efficacy while maintaining computational efficiency.

**Effect of Number of Fusion Tokens.** The number of fusion tokens $K$ in the LLM-driven multidimensional representation fusion module is a crucial hyperparameter that influences the model's capacity to capture diverse evaluation dimensions. We conduct ablation experiments by varying $K$ from 2 to 6, with results summarized in Table 6. As observed, performance improves as $K$ increases from 2 to 4, indicating that a higher number of fusion tokens enhances the model's ability to encode multifaceted semantics. However, beyond $K = 4$, gains plateau and even slightly decline at $K = 6$, likely due to over-parameterization leading to redundancy and overfitting. Therefore, we select $K = 4$ as the optimal setting, balancing expressiveness and generalization.

**Effect of Dual-Path Modules in MDA.** To mitigate potential hallucinations arising from modality missingness during LLM inference, we introduce the Mask-Aware Dual-Path Aggregation (MDA) mechanism. MDA comprises two distinct pathways: Path 1 (Uncertainty-Calibrated Reasoning), which focuses on high-level semantic inference, and Path 2 (Cross-Modal

*Table 8.* Ablation study on LoRA hyperparameters (Rank $r$ and Alpha $\alpha$) on the FS1000 and Fis-V datasets.

| Rank ($r$) | Alpha ($\alpha$) | FS1000 | | Fis-V | |
|---|---|---|---|---|---|
| | | Average | $\{v, f, a\}$ | Average | $\{v, f, a\}$ |
| 8 | 16 | 0.772 / 16.32 | 0.880 / 12.45 | 0.729 / 17.55 | 0.830 / 12.80 |
| 8 | 32 | 0.778 / 15.10 | 0.885 / 11.60 | 0.735 / 16.40 | 0.836 / 11.20 |
| 16 | 16 | 0.781 / 14.85 | 0.887 / 11.25 | 0.738 / 15.95 | 0.838 / 10.95 |
| **16** | **32** | **0.789 / 14.08** | **0.891 / 10.44** | **0.743 / 15.09** | **0.841 / 10.19** |
| 16 | 64 | 0.785 / 14.42 | 0.889 / 11.05 | 0.740 / 15.65 | 0.839 / 10.55 |
| 32 | 32 | 0.783 / 14.65 | 0.888 / 11.30 | 0.741 / 15.80 | 0.840 / 10.85 |
| 32 | 64 | 0.780 / 15.20 | 0.886 / 11.95 | 0.736 / 16.90 | 0.835 / 11.50 |

*Table 9.* Ablation study on prompt designs. "w/o Condition": Removing the explicit textual description of modality availability in PCMI; "w/o Task Inst.": Removing the core task instructions in PCMI that guide the LLM to infer missing data; "w/o Fusion Guide": Removing the instruction in LMRF that prompts the LLM to integrate features into the special tokens; "w/o Special Tokens": Removing all special boundary and fusion tokens, feeding only raw features and text, then pooling the last hidden state; and "w/o All Prompts": Removing all textual instructions and special tokens, reducing the LLM to a standard feature encoder.

| Settings | FS1000 | | Fis-V | |
|---|---|---|---|---|
| | Average | $\{v, f, a\}$ | Average | $\{v, f, a\}$ |
| w/o Condition | 0.774 / 15.62 | 0.880 / 12.35 | 0.730 / 16.52 | 0.832 / 11.45 |
| w/o Task Inst. | 0.768 / 16.05 | 0.875 / 13.10 | 0.725 / 17.20 | 0.825 / 12.15 |
| w/o Fusion Guide | 0.778 / 15.10 | 0.884 / 11.55 | 0.735 / 16.10 | 0.835 / 11.02 |
| w/o Special Tokens | 0.755 / 18.32 | 0.865 / 14.80 | 0.715 / 18.55 | 0.812 / 13.50 |
| w/o All Prompts | 0.710 / 22.45 | 0.830 / 18.25 | 0.675 / 22.10 | 0.785 / 16.45 |
| **Full Model** | **0.789 / 14.08** | **0.891 / 10.44** | **0.743 / 15.09** | **0.841 / 10.19** |

Pattern Recovery), which emphasizes low-level feature aggregation and pattern recovery. We evaluate the contribution of each mask-aware path through ablation studies, as detailed in Table 7. Removing the MDA mechanism entirely, *i.e.*, simply averaging the outputs of the two paths without calibration, results in a significant performance decline. Furthermore, while utilizing either Path 1 or Path 2 independently yields performance gains, the complete MDA mechanism synergizing both pathways achieves the best results. This confirms the effectiveness of MDA in mitigating hallucinations and enhancing model robustness against incomplete observations.

**Effect of LoRA Configurations.** To ensure parameter efficiency within our LIMSSR framework, we employ Low-Rank Adaptation (LoRA) (Hu et al., 2022) for fine-tuning the LLM backbone. We conduct ablation studies to investigate the impact of LoRA hyperparameters, specifically the rank $r$ and the scaling factor $\alpha$. As detailed in Table 8, we evaluate various combinations of rank values ($r \in \{8, 16, 32\}$) and scaling factors ($\alpha \in \{16, 32, 64\}$) on the FS1000 and Fis-V datasets. We observe consistent improvements when increasing $r$ from 8 to 16, while further enlarging to 32 brings no additional gains, suggesting diminishing returns with larger ranks. Similarly, $\alpha = 32$ achieves the most stable performance across settings. Overall, $r = 16$ and $\alpha = 32$ yieldsthe best results on both datasets, offering a favorable trade-off between performance and parameter efficiency. Consequently, we adopt this configuration for all subsequent experiments.

**Effect of Prompt Designs.** Our LIMSSR relies on two core prompt-driven mechanisms: Prompt-Guided Context-Aware Modality Imputation (PCMI) and LLM-Driven Multidimensional Representation Fusion (LMRF). To validate the efficacy of our prompt engineering, we conduct an ablation study on different prompt components, as detailed in Table 9. We define five variants: 1) "w/o Condition": Removing the explicit textual description of modality availability in PCMI; 2) "w/o Task Inst.": Removing the core task instructions in PCMI that guide the LLM to infer missing data; 3) "w/o Fusion Guide": Removing the instruction in LMRF that prompts the LLM to integrate features into the special tokens; 4) "w/o Special Tokens": Removing all special boundary and fusion tokens, feeding only raw features and text, then pooling the last hidden state; and 5) "w/o All Prompts": Removing all textual instructions and special tokens, reducing the LLM to a standard feature encoder. The results indicate that removing any component degrades performance. Notably, the exclusion of "Task

*Table 10.* Ablation study on different prompt formulations on the FS1000 and Fis-V datasets.

| Prompt | FS1000 | | Fis-V | |
|---|---|---|---|---|
| | Average | $\{v, f, a\}$ | Average | $\{v, f, a\}$ |
| Template 1 (Ours) | 0.789 / **14.08** | **0.891** / 10.44 | 0.743 / 15.09 | 0.841 / 10.19 |
| Template 2 | 0.786 / 14.22 | 0.889 / 10.61 | 0.739 / 15.23 | 0.838 / 10.49 |
| Template 3 | **0.792** / 14.10 | 0.890 / **10.31** | **0.747 / 14.98** | **0.844 / 10.05** |
| Template 4 | 0.787 / 14.28 | 0.890 / 10.55 | 0.741 / 15.30 | 0.839 / 10.32 |

Inst." causes the most significant drop among partial removals, underscoring the critical role of explicit task definition in guiding the LLM's reasoning process. Furthermore, the substantial performance gap between the full model and "w/o All Prompts" confirms that our prompt design successfully unlocks the LLM's capability to handle incomplete multimodal data, rather than merely using it as a large backbone.

**Prompt Templates for Robustness Analysis.** To evaluate prompt formulation robustness, we use four semantically equivalent templates with different surface structures. In all templates, {avail.} and {miss.} are instantiated according to the current modality condition. As shown in Table 10, LIMSSR remains stable across structured, reasoning-oriented, and multi-step prompt formulations. Across both datasets and both incomplete/complete settings, the maximum fluctuation relative to the original template is only 0.004 in $\rho$ and 0.30 in MSE. The reasoning-oriented template provides slight gains, while the structured and multi-step templates remain comparable to the original design. These results indicate that LIMSSR benefits from the semantic role of the prompt rather than being sensitive to surface-level wording.

**Template 1 (Original).**

```
Given the available {avail.} features from an action video. The {miss.} modality is
    missing. Based on the available modalities, please infer and reconstruct the useful
    latent representations for the missing {miss.} modalities at the designated positions.
     Then integrate and enhance all multimodal features for action quality assessment.
    Output the fused multi-dimensional feature representations at the designated feature
    dimension positions:
```

**Template 2 (Structured).**

```
Available Modalities: {avail.}
Missing Modalities: {miss.}

Task:
1. Infer the latent representations of the missing modalities based on available inputs.
2. Reconstruct the missing modality features at the designated positions.
3. Fuse all modalities to enhance feature representations for action quality assessment.

Output:
Output the fused feature representations at the designated feature dimension positions.
```

**Template 3 (Reasoning-Oriented).**

```
Given the available modalities {avail.} and missing modalities {miss.}, reason about the
    relationships between modalities. First, infer the latent representations of the
    missing modalities based on cross-modal dependencies. Then reconstruct the missing
    features and integrate them with available ones. Finally, produce enhanced multimodal
    representations for action quality assessment.

Output the fused feature representations at the designated feature dimension positions.
```

**Template 4 (Multi-step Explicit).**

```
You are given multimodal features from an action video.

Step 1: Identify the available modalities: {avail.} and missing modalities: {miss.}
Step 2: Estimate the latent representations of the missing modalities.
```

*Table 11.* Compare the computational costs with the state-of-the-art methods on the FS1000. **T-Miss**: Training-time missing modality.

| Methods | T-Miss | #Params | #FLOPs | Average | $\{v, f, a\}$ |
|---|---|---|---|---|---|
| ActionMAE (AAAI'23) | ✗ | 14.05 M | 62.12 G | 0.651 / 38.18 | 0.809 / 17.96 |
| GCNet (TPAMI'23) | ✗ | 8.78 M | 1191.39 G | 0.610 / 28.62 | 0.764 / 21.82 |
| IMDer (NeurIPS'23) | ✗ | 7.97 M | 23.53 G | 0.636 / 27.86 | 0.788 / 25.95 |
| MLP-Mixer (AAAI'23) | ✗ | 14.32 M | 49.90 G | 0.520 / 71.78 | 0.819 / 14.56 |
| PAMFN (TIP'24) | ✗ | 18.06 M | 2.56 G | 0.571 / 75.16 | 0.855 / 13.02 |
| MoMKE (ACMMM'24) | ✗ | 5.39 M | 2.60 G | 0.668 / 26.08 | 0.819 / 16.85 |
| SDR-GNN (KBS'25) | ✗ | 22.63 M | 24.35 G | 0.665 / 24.79 | 0.817 / 15.91 |
| MCMoE (AAAI'26) | ✗ | 4.90 M | 1.34 G | 0.782 / 15.37 | 0.881 / 11.53 |
| **LIMSSR** (This Paper) | ✓ | 600.38 M (Qwen3-0.6B) + 17.35 M | 232.966 G (Qwen3-0.6B) + 0.57 G | 0.789 / 14.08 | 0.891 / 10.44 |

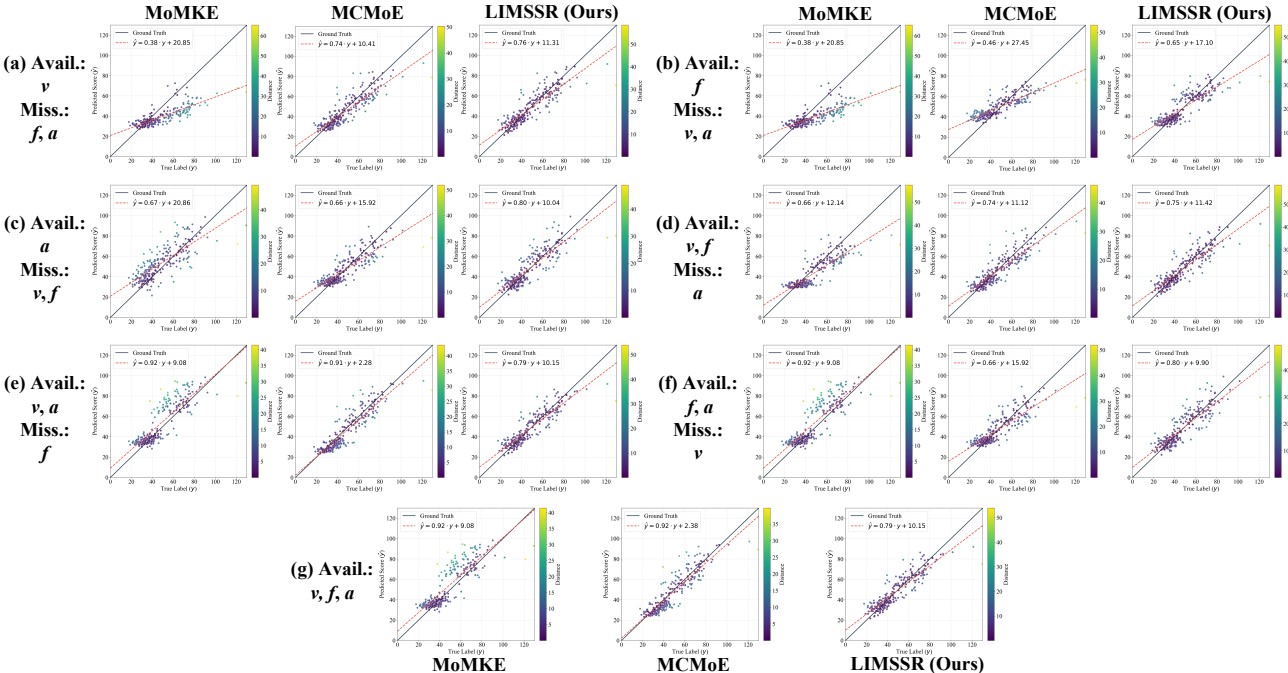

*Figure 4.* Comparative scatter plots illustrating the prediction fidelity of our approach against state-of-the-art baselines, MCMoE (Xu et al., 2026) (incomplete multimodal AQA) and MoMKE (Xu et al., 2024d) (incomplete multimodal emotion recognition), across all modality combinations on the FS1000 (TES) benchmark. Each point maps the ground-truth score (x-axis) to the predicted score (y-axis), with color intensity denoting the magnitude of the prediction error.

```
Step 3: Reconstruct missing modality features at the designated positions.
Step 4: Fuse all modalities into a unified representation for action quality assessment.

Output the fused feature representations at the designated feature dimension positions.
```

**Computational Cost Analysis.** In Table 11, we benchmark the computational efficiency of LIMSSR against state-of-the-art methods on the FS1000 dataset. The input for all methods is uniformly three-modal features containing information from 95 video clips. While LIMSSR leverages an LLM backbone, we strategically select the edge-deployable, sub-1B parameter model (Qwen3-0.6B) to ensure computational feasibility while maintaining scalability for larger architectures. Excluding the pre-trained backbone, our framework introduces minimal overhead, adding only approximately 17.35M parameters and 0.57 GFLOPs. This marginal increase is justified by the significant performance gains, demonstrating a favorable trade-off between efficiency and accuracy. Furthermore, by employing Parameter-Efficient Fine-Tuning (PEFT) via LoRA, we only update 159.91M parameters within the LLM during training. The total number of trainable parameters is strictly limited to 177.26M, enabling cost-effective optimization. Consequently, our model can be trained on a single consumer-grade NVIDIA

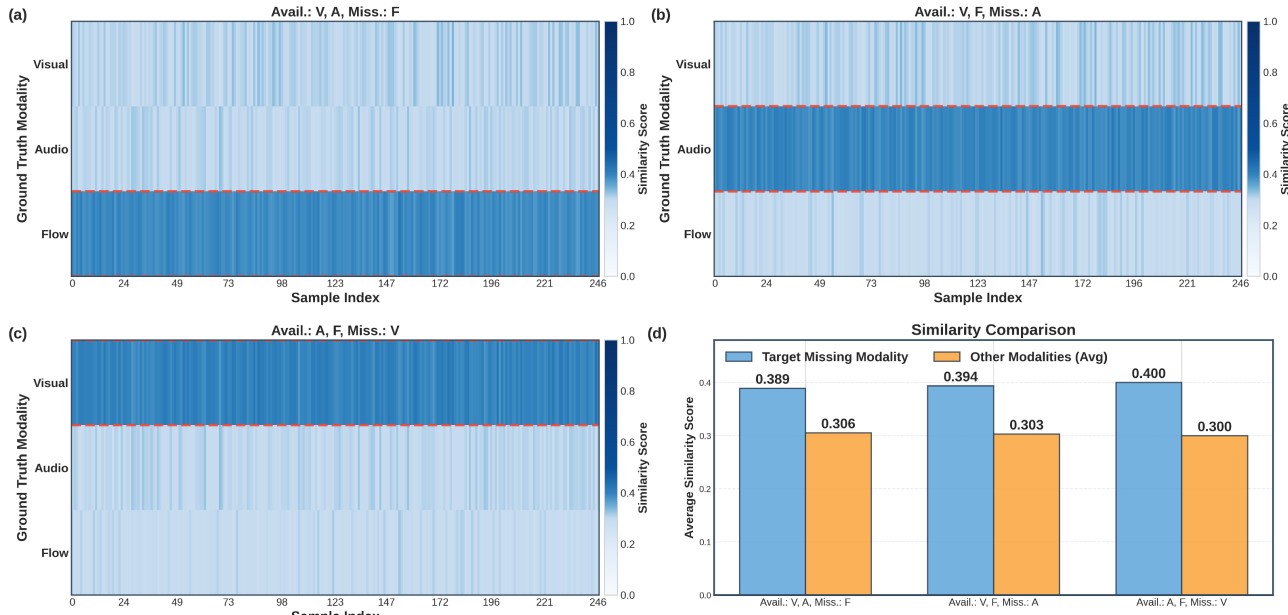

*Figure 5.* Visualization of the similarity between the semantic representations inferred by LIMSSR and the ground-truth modality representations on the FS1000 test set (TES). (a-c) Display the cosine similarity between the latent semantics inferred by the LLM and the ground-truth representations of all three modalities when the missing modality is Flow, Audio, and Visual, respectively. Each column represents a sample instance, and each row corresponds to a modality. The similarities within each column are normalized to sum to 1. The red dashed box indicates the target missing modality. (d) A bar chart comparison of the similarity between the inferred representation and the target missing modality versus other modalities across the three missingness scenarios. The results indicate that the inferred representations exhibit consistently higher similarity to the target missing modalities, demonstrating that LIMSSR effectively leverages available context to infer useful latent semantics for compensating information loss.

RTX 3090 GPU, significantly lowering the barrier for deployment in resource-constrained environments.

## D. More Visualizations

In this section, we present additional visualizations to substantiate the efficacy of our proposed framework in handling incomplete multimodal learning and action quality assessment.

**Visualization of Predicted Scores.** To qualitatively validate assessment fidelity, we present scatter plots of predicted versus ground-truth scores in Figure 4, benchmarking LIMSSR against state-of-the-art incomplete multimodal methods, MoMKE (Xu et al., 2024d) and MCMoE (Xu et al., 2026). Each point maps the ground-truth score (x-axis) to the predicted score (y-axis), with color intensity denoting the magnitude of the prediction error. The visible alignment of data points along the diagonal indicates that our framework achieves significantly tighter correlation and reduced estimation variance compared to baselines. Crucially, unlike competing methods that rely on the assumption of complete modalities during training, LIMSSR secures superior predictive performance even under *Training-Time Incomplete Observations*, underscoring its exceptional robustness and efficacy in handling severe modality missingness.

**Effectiveness of LLM-Driven Latent Inference.** A core premise of LIMSSR is that LLMs can act as reasoning engines to infer missing semantics from available context, rather than performing low-level pixel reconstruction. To validate whether our framework truly recovers meaningful representations for missing modalities, we compute the cosine similarity between the latent semantics inferred by the LLM and the ground-truth representations of the all modalities on the FS1000 test set (TES). Figure 5 (a-c) visualize these similarity distributions for missing Flow, Audio, and Visual scenarios, respectively. Each column represents a test sample, and darker colors indicate higher similarity. Each row corresponds to a modality. The similarities within each column are normalized to sum to 1. As observed, the inferred latent states consistently exhibit the highest similarity to their corresponding ground-truth missing modalities compared to non-corresponding modalities. For quantitative precision, Figure 5 (d) aggregates these results, showing that the inferred representations maintain a significantly higher affinity with the target ground truth. This empirical evidence confirms that LIMSSR successfully

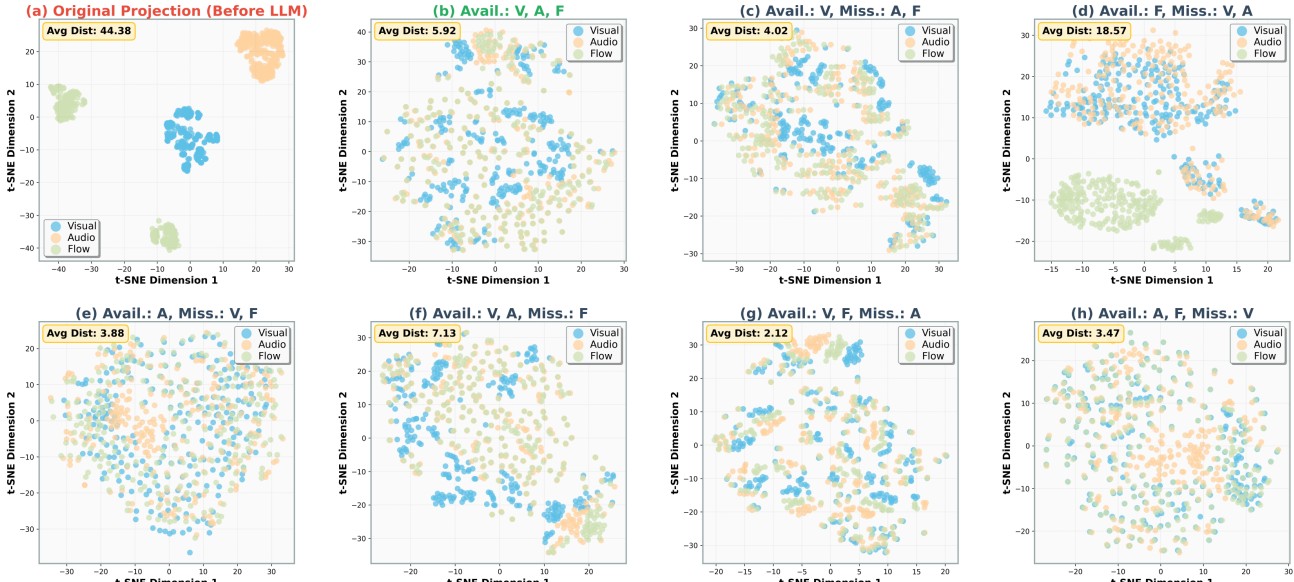

*Figure 6.* t-SNE visualization of multimodal representations on FS1000 test set (PCS). (a) depicts the raw features without LLM processing, while (b-h) show the representations after LLM-driven inference under various incomplete modality combinations. The average distances between modality clusters are annotated. After reasoning, the inferred representations for missing modalities align closely with the available ones, effectively bridging the cross-modal semantic gap.

leverages cross-modal correlations to hallucinate contextually accurate semantics, effectively compensating for information loss in the latent space.

**The t-SNE Visualizations of LLM-Driven Reasoning.** Our LIMSSR framework reformulates incomplete multimodal learning as a conditional sequence reasoning task, harnessing the sophisticated inferential capabilities of Large Language Models (LLMs). Rather than enforcing pixel-level reconstruction of absent features, our objective is to prompt the LLM to infer latent semantics from the available context, thereby mitigating cross-modal discrepancies and compensating for missing information. Figure 6 illustrates the t-SNE visualizations of feature representations for the three modalities on the FS1000 (PCS) dataset. Compared to the raw representations without LLM processing (Figure 6(a)), the representations inferred by the LLM across various incomplete scenarios (Figure 6(b-h)) demonstrate significantly improved alignment. Specifically, the latent representations generated for missing modalities are implicitly aligned with the available modalities in the semantic space. This convergence suggests that LIMSSR effectively utilizes the reasoning priors of LLMs to synthesize valuable latent semantics regardless of the missingness pattern, bridging the information gap and enhancing cross-modal synergy for robust quality assessment. Similarly, we present visualization results on the Fis-V (TES) and RG (Ribbon) datasets in Figure 7 and Figure 8, further verifying the generalizability and effectiveness of our method.

**Attention Visualization of Fusion Tokens.** While Figure 3 in the main text illustrates the attention distribution of the $K$ fusion tokens on the FS1000 dataset, we provide comprehensive visualizations here to further substantiate their role in multidimensional evaluation. Specifically, Figure 9, Figure 10, and Figure 11 depict the attention heatmaps of these $K$ fusion tokens across diverse incomplete modality combinations on the FS1000, Fis-V, and RG datasets, respectively. We observe that distinct tokens exhibit significantly different attention patterns depending on the missingness configuration, suggesting they play complementary roles in capturing diverse evaluation dimensions. This dynamic adjustment mechanism enables LIMSSR to flexibly adapt to various incomplete input scenarios, thereby enhancing overall assessment performance. Crucially, unlike existing approaches that pursue pixel-level reconstruction, our method reformulates incomplete multimodal learning as a conditional sequence reasoning task. By leveraging the powerful reasoning capabilities of LLMs to infer latent semantic representations for missing modalities, we effectively bridge cross-modal discrepancies and compensate for information loss. Consequently, benefiting from the next-token prediction mechanism of LLMs, salient information tends to aggregate within specific key tokens. As evidenced in these visualizations, even when modalities are missing, our model dynamically re-allocates attention to focus on information-dense and semantically rich tokens, facilitating more effective multimodal fusion and reasoning.

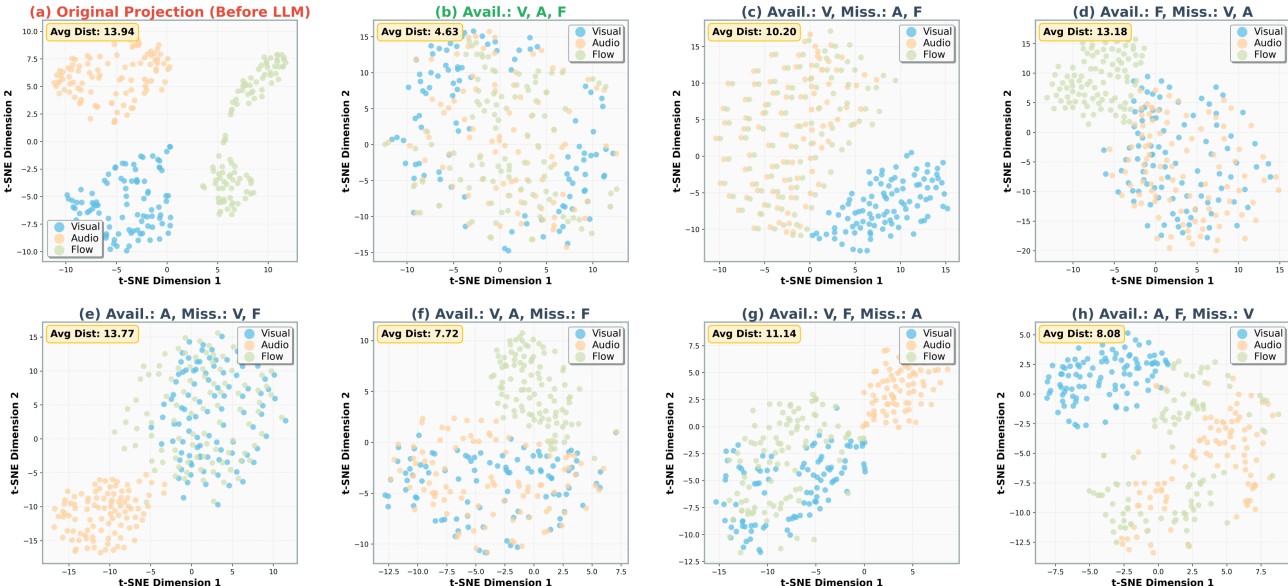

*Figure 7.* t-SNE visualization of multimodal representations on Fis-V test set (TES). (a) depicts the raw features without LLM processing, while (b-h) show the representations after LLM-driven inference under various incomplete modality combinations. The average distances between modality clusters are annotated. After reasoning, the inferred representations for missing modalities align closely with the available ones, effectively bridging the cross-modal semantic gap.

## E. Limitations and Broader Impacts

**Limitations.** While our LIMSSR framework demonstrates robust performance under training-time incomplete observations, several limitations persist. First, although LLMs possess strong reasoning capabilities, fine-tuning them for challenging downstream tasks often requires sufficient data to fully unlock their potential. Specifically, current Action Quality Assessment (AQA) scenarios demand high fine-grained understanding; consequently, on smaller datasets like Rhythmic Gymnastics (RG), the performance gains are constrained by limited sample availability. Future work will explore data-efficient fine-tuning strategies to improve utilization. Second, while we instantiate our approach on the challenging task of long-term AQA, the framework is designed to be generalizable. We plan to investigate its applicability to other multimodal domains in future research. Finally, although the proposed Mask-Aware Dual-Path Aggregation (MDA) mitigates hallucinations, inference reliability in extreme missingness scenarios can be further improved. Future iterations may incorporate physical constraints to enhance reasoning robustness.

**Broader Impacts.** This work reformulates incomplete multimodal learning as a conditional sequence reasoning task, introducing the LIMSSR framework to address the challenging scenario where modalities are missing during both training and testing. By leveraging the reasoning capabilities and prior world knowledge of pre-trained Large Language Models (LLMs), we effectively tackle the issue of training-time incomplete observations. Our method adopts a sub-1B parameter LLM backbone, ensuring feasibility for edge deployment while retaining scalability for larger models. Furthermore, through instantiation in incomplete multimodal AQA, this research significantly enhances model accuracy and real-world applicability, with potential benefits for sports scoring, healthcare rehabilitation, and skill assessment. However, it is crucial to note that AQA models, including ours, are intended to provide objective references and should not serve as the sole criterion for evaluation, as no model guarantees perfect accuracy.

## F. Extended Comparison with State-of-the-Art Methods

Following established protocols in prior work (Xu et al., 2026), we conduct comprehensive evaluations on three long-term Action Quality Assessment (AQA) benchmarks (FS1000, Fis-V, and Rhythmic Gymnastics), covering both incomplete and complete multimodal scenarios. In the main manuscript, due to space constraints, we primarily report the aggregated average metrics across all action classes and scoring categories for each benchmark.

In this section, we provide a granular breakdown of the performance comparisons. Table 12, Table 13, and Table 14 detail

the results for the FS1000, Fis-V, and Rhythmic Gymnastics (RG) datasets, respectively. These tables demonstrate that LIMSSR consistently maintains superior or highly competitive performance across diverse missingness patterns and specific scoring criteria, further validating its robustness under training-time incomplete observations.

***Detailed experimental results and more visualizations are presented in the subsequent pages.***

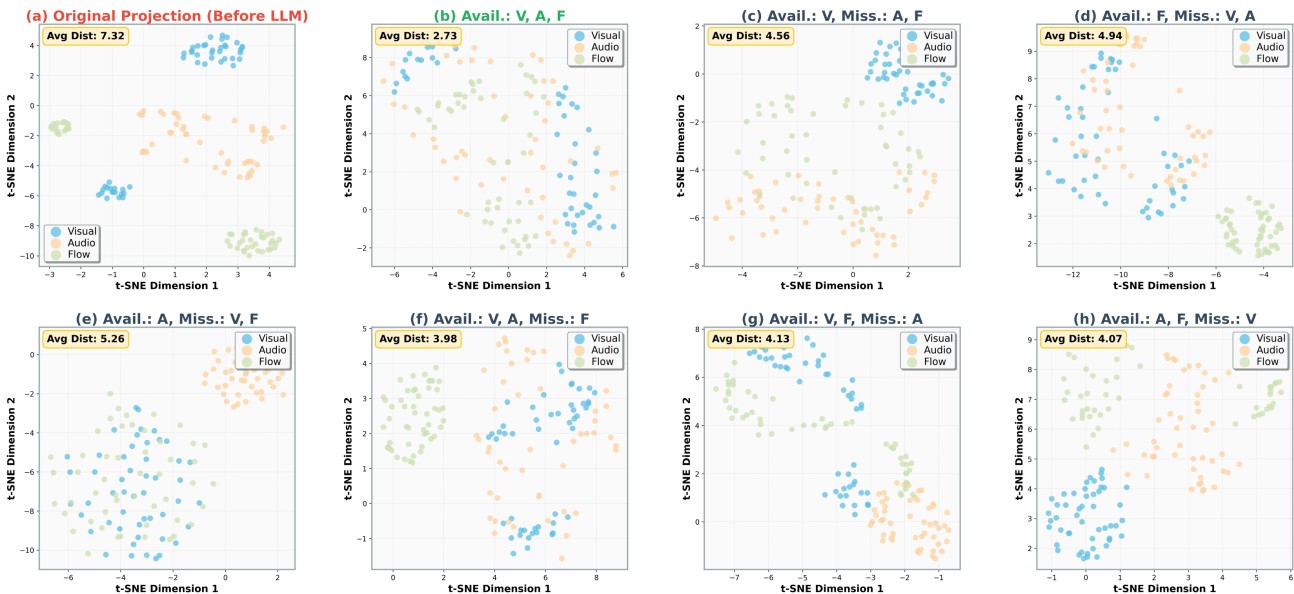

*Figure 8.* t-SNE visualization of multimodal representations on RG test set (Ribbon). (a) depicts the raw features without LLM processing, while (b-h) show the representations after LLM-driven inference under various incomplete modality combinations. The average distances between modality clusters are annotated. After reasoning, the inferred representations for missing modalities align closely with the available ones, effectively bridging the cross-modal semantic gap.

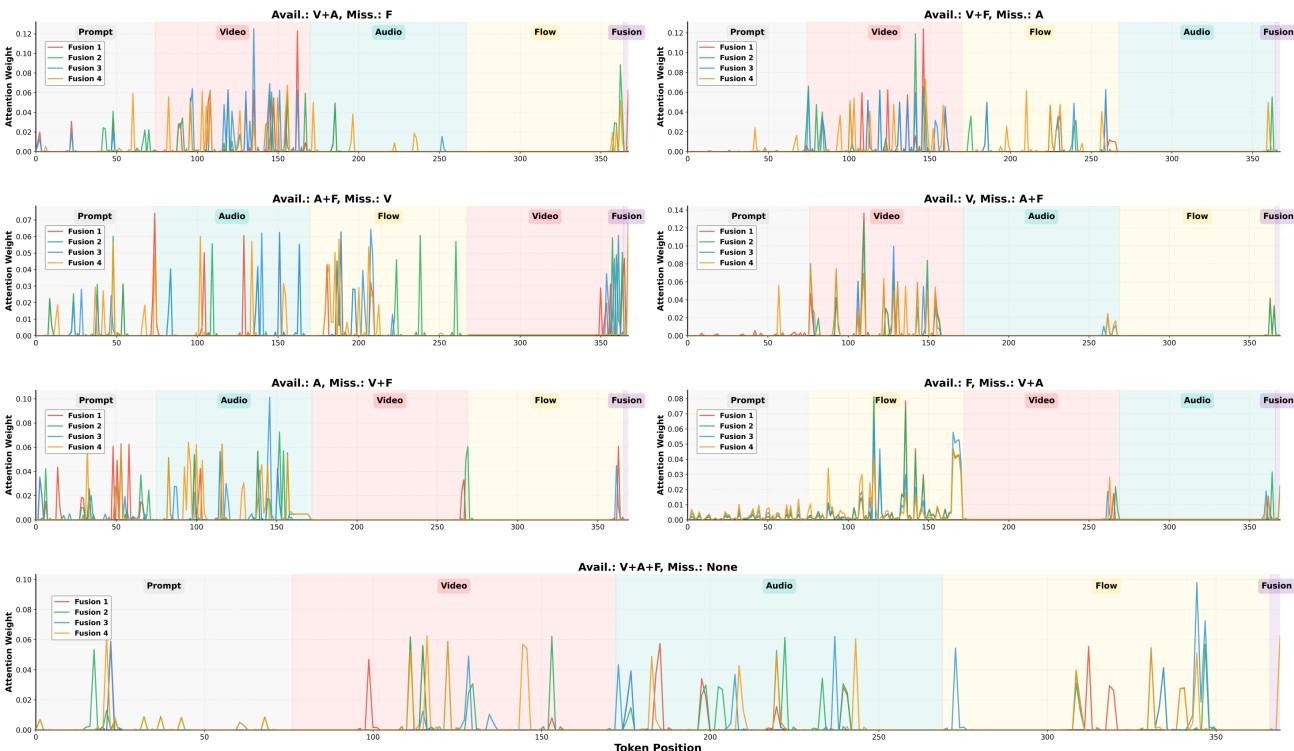

*Figure 9.* Visualization of fusion token attention weights across different modality combinations on the FS1000 (PCS).

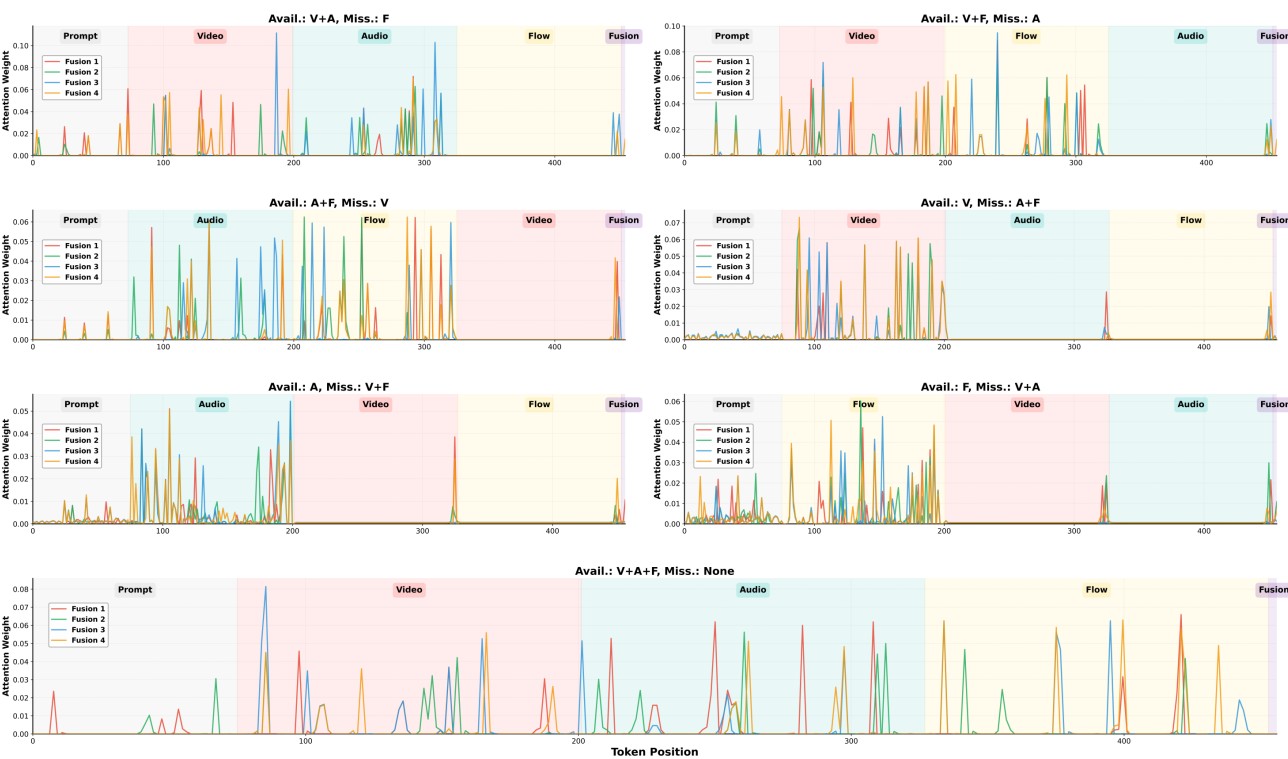

*Figure 10.* Visualization of fusion token attention weights across different modality combinations on the Fis-V (TES).

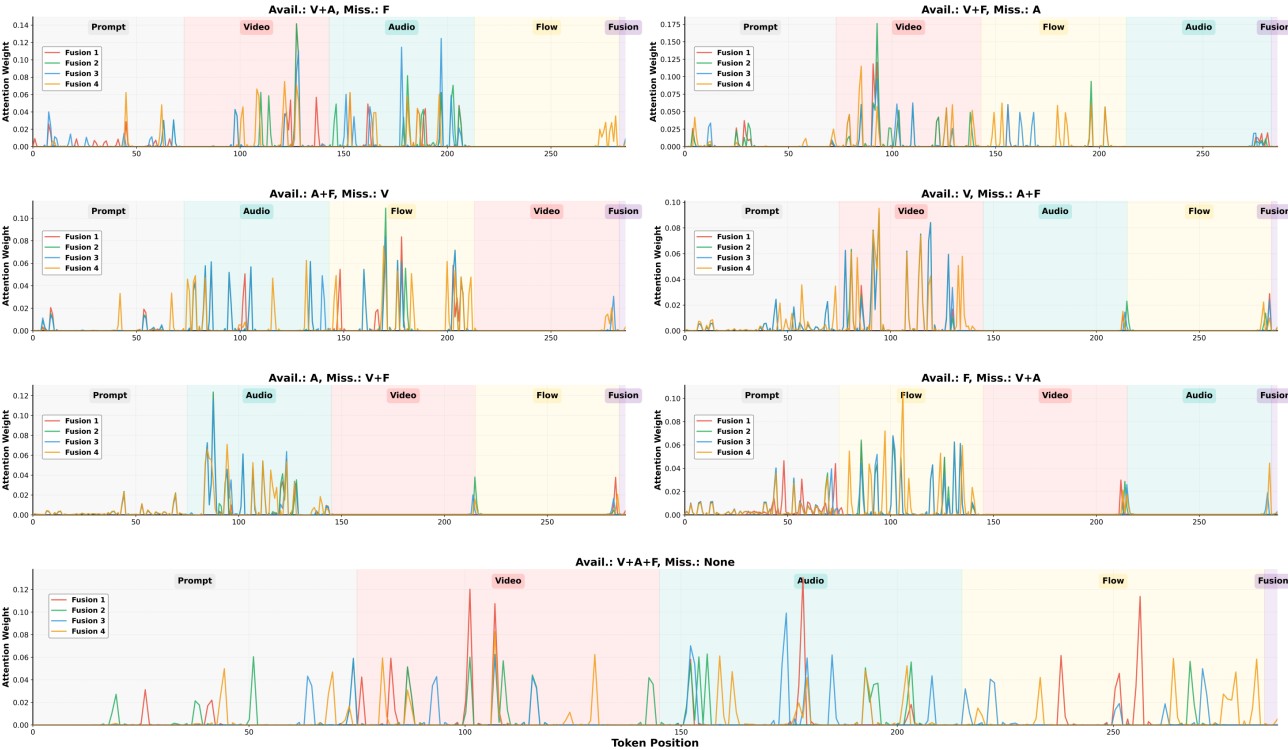

*Figure 11.* Visualization of fusion token attention weights across different modality combinations on the RG (Ribbon).

*Table 12.* Comparisons of performance on the FS1000 under incomplete multimodal scenarios. $v$, $f$, and $a$ refer to the RGB, flow, and audio modalities. "Average" denotes the average result of all six incomplete multimodal combinations. The **bold** / underline indicate the best / second-best results. ♡, ♣, and ♠ mean the evaluated method sources for incomplete/complete multimodal AQA, incomplete multimodal action recognition, and incomplete multimodal emotion recognition. **T-Miss**: Training-time missing modality. $\Delta_{SOTA}$ means the performance increase or decrease of our LIMSSR compared to the best competing methods.

| Types | Methods | T-Miss | Testing Condition (Spearman Correlation (↑) / Mean Squared Error (↓)) | | | | | | | |
| --- | --- | --- | --- | --- | --- | --- | --- | --- | --- | --- |
| | | | $\{v,f\}$ | $\{v,a\}$ | $\{f,a\}$ | $\{v\}$ | $\{f\}$ | $\{a\}$ | Average | $\{v,f,a\}$ |
| TES | ♣ActionMAE (AAAI'23) | ✗ | 0.853 / 144.66 | 0.833 / 410.45 | 0.862 / 138.01 | 0.800 / 328.56 | 0.778 / 118.02 | 0.762 / 256.94 | 0.818 / 232.77 | 0.881 / 107.07 |
| | ♠GCNet (TPAMI'23) | ✗ | 0.813 / 152.23 | 0.776 / 142.94 | 0.801 / 150.07 | 0.763 / 152.35 | 0.785 / 183.14 | 0.741 / 233.26 | 0.781 / 167.00 | 0.820 / 129.12 |
| | ♠IMDer (NeurIPS'23) | ✗ | 0.847 / 122.66 | 0.827 / 179.14 | 0.833 / 137.80 | 0.737 / 213.77 | 0.742 / 123.89 | 0.805 / 194.01 | 0.802 / 161.88 | 0.853 / 159.36 |
| | ♡MLP-Mixer (AAAI'23) | ✗ | 0.749 / 156.74 | 0.366 / 380.63 | 0.015 / 573.44 | 0.286 / 675.68 | 0.276 / 578.91 | 0.457 / 429.42 | 0.386 / 465.80 | 0.877 / 86.54 |
| | ♡PAMFN (TIP'24) | ✗ | 0.827 / 388.21 | 0.351 / 397.58 | 0.434 / 360.58 | 0.677 / 607.79 | 0.011 / 788.74 | 0.241 / 372.50 | 0.474 / 485.90 | 0.874 / 78.42 |
| | ♠MoMKE (ACMMM'24) | ✗ | 0.843 / 108.53 | 0.853 / 146.50 | 0.854 / 141.05 | 0.813 / 234.39 | 0.788 / 158.86 | 0.798 / 152.56 | 0.827 / 152.56 | 0.876 / 97.97 |
| | ♠SDR-GNN (KBS'25) | ✗ | 0.861 / 98.49 | 0.807 / 153.41 | 0.856 / 120.34 | 0.810 / 164.88 | 0.811 / 177.01 | 0.778 / 141.63 | 0.823 / 142.63 | 0.872 / 88.89 |
| | ♡MCMoE (AAAI'26) | ✗ | 0.883 / 77.76 | 0.901 / 73.64 | 0.878 / 81.65 | 0.879 / 84.11 | 0.835 / 129.01 | 0.858 / 91.90 | 0.874 / 89.68 | 0.900 / 71.69 |
| | LIMSSR (This Paper) | ✓ | 0.887 / 77.44 | 0.907 / 65.25 | 0.867 / 85.80 | 0.889 / 77.52 | 0.842 / 95.53 | 0.863 / 87.71 | 0.878 / 81.54 | 0.907 / 64.58 |
| | $\Delta_{SOTA}$ | - | ↑0.5% / ↓0.4% | ↑0.7% / ↓11.4% | ↓1.3% / ↑5.1% | ↑1.1% / ↓7.8% | ↑0.8% / ↓19.1% | ↑0.6% / ↓4.6% | ↑0.5% / ↓9.1% | ↑0.8% / ↓9.9% |
| PCS | ♣ActionMAE (AAAI'23) | ✗ | 0.734 / 25.13 | 0.530 / 35.87 | 0.613 / 42.83 | 0.745 / 23.18 | 0.440 / 26.44 | 0.314 / 28.72 | 0.583 / 30.36 | 0.784 / 16.04 |
| | ♠GCNet (TPAMI'23) | ✗ | 0.674 / 21.87 | 0.691 / 20.68 | 0.477 / 28.16 | 0.675 / 26.05 | 0.476 / 28.68 | 0.401 / 34.32 | 0.577 / 26.63 | 0.692 / 20.61 |
| | ♠IMDer (NeurIPS'23) | ✗ | 0.688 / 28.16 | 0.602 / 16.55 | 0.608 / 30.15 | 0.594 / 34.39 | 0.592 / 29.67 | 0.466 / 27.89 | 0.595 / 27.80 | 0.725 / 19.53 |
| | ♡MLP-Mixer (AAAI'23) | ✗ | 0.714 / 17.95 | 0.571 / 32.46 | 0.609 / 26.25 | 0.717 / 24.71 | 0.595 / 26.09 | 0.005 / 37.12 | 0.565 / 27.43 | 0.799 / 12.47 |
| | ♡PAMFN (TIP'24) | ✗ | 0.620 / 25.49 | 0.496 / 26.08 | 0.612 / 26.54 | 0.722 / 24.39 | 0.618 / 24.98 | 0.166 / 42.26 | 0.558 / 28.29 | 0.804 / 13.72 |
| | ♠MoMKE (ACMMM'24) | ✗ | 0.784 / 20.65 | 0.791 / 17.68 | 0.528 / 27.83 | 0.745 / 27.95 | 0.314 / 30.86 | 0.481 / 28.28 | 0.638 / 25.54 | 0.800 / 17.61 |
| | ♠SDR-GNN (KBS'25) | ✗ | 0.794 / 20.83 | 0.773 / 19.07 | 0.614 / 28.77 | 0.694 / 30.18 | 0.491 / 26.84 | 0.479 / 30.43 | 0.658 / 26.02 | 0.801 / 19.73 |
| | ♡MCMoE (AAAI'26) | ✗ | 0.836 / 8.92 | 0.866 / 7.72 | 0.699 / 18.99 | 0.830 / 9.49 | 0.619 / 24.03 | 0.584 / 20.66 | 0.759 / 14.97 | 0.872 / 7.43 |
| | LIMSSR (This Paper) | ✓ | 0.845 / 8.21 | 0.891 / 6.90 | 0.678 / 15.84 | 0.848 / 8.05 | 0.574 / 28.33 | 0.644 / 17.38 | 0.772 / 14.12 | 0.892 / 6.84 |
| | $\Delta_{SOTA}$ | - | ↑1.1% / ↓8.0% | ↑2.9% / ↓10.6% | ↓3.0% / ↓16.6% | ↑2.2% / ↓15.2% | ↓7.3% / ↑17.9% | ↑10.3% / ↓15.9% | ↑1.7% / ↓5.7% | ↑2.3% / ↓7.9% |
| SS | ♣ActionMAE (AAAI'23) | ✗ | 0.694 / 0.52 | 0.750 / 0.47 | 0.474 / 0.88 | 0.689 / 0.55 | 0.491 / 1.06 | 0.449 / 0.92 | 0.605 / 0.73 | 0.788 / 0.46 |
| | ♠GCNet (TPAMI'23) | ✗ | 0.723 / 1.45 | 0.577 / 0.87 | 0.449 / 1.76 | 0.670 / 1.15 | 0.304 / 1.11 | 0.322 / 1.46 | 0.527 / 1.30 | 0.727 / 0.61 |
| | ♠IMDer (NeurIPS'23) | ✗ | 0.761 / 2.48 | 0.640 / 0.82 | 0.548 / 2.20 | 0.752 / 0.54 | 0.279 / 2.40 | 0.282 / 1.23 | 0.575 / 1.61 | 0.776 / 0.53 |
| | ♡MLP-Mixer (AAAI'23) | ✗ | 0.711 / 0.64 | 0.313 / 3.59 | 0.678 / 0.88 | 0.545 / 2.39 | 0.530 / 0.96 | 0.041 / 3.97 | 0.498 / 2.07 | 0.803 / 0.68 |
| | ♡PAMFN (TIP'24) | ✗ | 0.533 / 0.82 | 0.664 / 11.80 | 0.624 / 0.74 | 0.746 / 5.45 | 0.448 / 0.91 | 0.040 / 12.31 | 0.538 / 5.34 | 0.758 / 0.74 |
| | ♠MoMKE (ACMMM'24) | ✗ | 0.789 / 0.43 | 0.702 / 0.99 | 0.392 / 1.87 | 0.767 / 0.97 | 0.288 / 1.17 | 0.335 / 1.50 | 0.584 / 1.16 | 0.794 / 0.40 |
| | ♠SDR-GNN (KBS'25) | ✗ | 0.777 / 0.40 | 0.760 / 0.80 | 0.418 / 2.07 | 0.759 / 0.68 | 0.488 / 1.33 | 0.308 / 1.74 | 0.617 / 1.17 | 0.799 / 0.44 |
| | ♡MCMoE (AAAI'26) | ✗ | 0.844 / 0.32 | 0.876 / 0.27 | 0.687 / 0.63 | 0.841 / 0.31 | 0.563 / 0.76 | 0.541 / 0.82 | 0.755 / 0.52 | 0.876 / 0.26 |
| | LIMSSR (This Paper) | ✓ | 0.870 / 0.30 | 0.896 / 0.26 | 0.695 / 0.52 | 0.870 / 0.30 | 0.573 / 0.91 | 0.679 / 0.57 | 0.792 / 0.48 | 0.894 / 0.26 |
| | $\Delta_{SOTA}$ | - | ↑3.1% / ↓6.3% | ↑2.3% / ↓3.7% | ↑1.2% / ↓17.5% | ↑3.4% / ↓3.2% | ↑1.8% / ↑19.7% | ↑25.5% / ↓30.5% | ↑4.9% / ↓7.7% | ↑2.1% / ↓0.0% |
| TR | ♣ActionMAE (AAAI'23) | ✗ | 0.778 / 0.64 | 0.787 / 0.56 | 0.456 / 0.94 | 0.746 / 0.61 | 0.412 / 1.18 | 0.442 / 1.07 | 0.632 / 0.83 | 0.798 / 0.48 |
| | ♠GCNet (TPAMI'23) | ✗ | 0.661 / 0.92 | 0.742 / 0.77 | 0.394 / 1.97 | 0.691 / 0.80 | 0.293 / 1.76 | 0.459 / 1.02 | 0.562 / 1.21 | 0.748 / 0.46 |
| | ♠IMDer (NeurIPS'23) | ✗ | 0.759 / 0.76 | 0.797 / 0.53 | 0.492 / 0.87 | 0.760 / 0.59 | 0.308 / 1.02 | 0.461 / 0.92 | 0.629 / 0.78 | 0.792 / 0.44 |
| | ♡MLP-Mixer (AAAI'23) | ✗ | 0.689 / 0.89 | 0.252 / 1.11 | 0.578 / 1.65 | 0.590 / 0.91 | 0.555 / 1.19 | 0.056 / 1.40 | 0.478 / 1.19 | 0.805 / 0.38 |
| | ♡PAMFN (TIP'24) | ✗ | 0.615 / 1.25 | 0.565 / 1.08 | 0.545 / 1.18 | 0.677 / 1.04 | 0.600 / 0.86 | 0.116 / 1.32 | 0.537 / 1.12 | 0.810 / 0.62 |
| | ♠MoMKE (ACMMM'24) | ✗ | 0.779 / 0.45 | 0.799 / 0.42 | 0.443 / 1.01 | 0.781 / 0.45 | 0.302 / 1.27 | 0.413 / 0.99 | 0.626 / 0.77 | 0.795 / 0.41 |
| | ♠SDR-GNN (KBS'25) | ✗ | 0.774 / 0.83 | 0.758 / 0.67 | 0.414 / 1.61 | 0.728 / 0.76 | 0.323 / 1.08 | 0.445 / 0.99 | 0.604 / 0.79 | 0.807 / 0.57 |
| | ♡MCMoE (AAAI'26) | ✗ | 0.841 / 0.33 | 0.891 / 0.26 | 0.702 / 0.56 | 0.850 / 0.31 | 0.629 / 0.70 | 0.513 / 0.85 | 0.767 / 0.50 | 0.882 / 0.26 |
| | LIMSSR (This Paper) | ✓ | 0.858 / 0.32 | 0.885 / 0.27 | 0.671 / 0.61 | 0.856 / 0.33 | 0.527 / 0.75 | 0.633 / 0.68 | 0.769 / 0.49 | 0.887 / 0.27 |
| | $\Delta_{SOTA}$ | - | ↑2.0% / ↓3.0% | ↓0.7% / ↑3.8% | ↓4.4% / ↑8.9% | ↑0.7% / ↑6.5% | ↓16.2% / ↑7.1% | ↑23.4% / ↓20.0% | ↑0.3% / ↓2.0% | ↑0.6% / ↑3.8% |
| PE | ♣ActionMAE (AAAI'23) | ✗ | 0.784 / 0.64 | 0.781 / 0.72 | 0.378 / 1.19 | 0.783 / 0.65 | 0.292 / 1.34 | 0.255 / 2.02 | 0.595 / 1.09 | 0.790 / 0.72 |
| | ♠GCNet (TPAMI'23) | ✗ | 0.774 / 0.66 | 0.796 / 0.68 | 0.406 / 1.17 | 0.752 / 1.67 | 0.388 / 1.12 | 0.367 / 1.24 | 0.617 / 1.09 | 0.793 / 0.83 |
| | ♠IMDer (NeurIPS'23) | ✗ | 0.743 / 1.03 | 0.803 / 0.62 | 0.459 / 1.16 | 0.764 / 0.72 | 0.306 / 1.37 | 0.427 / 1.15 | 0.619 / 1.01 | 0.800 / 0.75 |
| | ♡MLP-Mixer (AAAI'23) | ✗ | 0.687 / 1.02 | 0.677 / 3.13 | 0.396 / 1.94 | 0.468 / 4.09 | 0.459 / 1.84 | 0.217 / 1.87 | 0.502 / 2.31 | 0.838 / 0.79 |
| | ♡PAMFN (TIP'24) | ✗ | 0.780 / 1.26 | 0.742 / 1.02 | 0.524 / 3.00 | 0.764 / 1.08 | 0.496 / 3.16 | 0.034 / 1.61 | 0.601 / 1.85 | 0.783 / 1.18 |
| | ♠MoMKE (ACMMM'24) | ✗ | 0.774 / 0.79 | 0.809 / 0.62 | 0.451 / 1.09 | 0.777 / 0.74 | 0.282 / 1.35 | 0.416 / 1.14 | 0.626 / 0.95 | 0.803 / 0.67 |
| | ♠SDR-GNN (KBS'25) | ✗ | 0.805 / 0.67 | 0.794 / 0.69 | 0.428 / 1.13 | 0.775 / 1.51 | 0.403 / 1.19 | 0.374 / 1.20 | 0.636 / 1.06 | 0.817 / 0.79 |
| | ♡MCMoE (AAAI'26) | ✗ | 0.828 / 0.59 | 0.871 / 0.52 | 0.703 / 0.94 | 0.830 / 0.59 | 0.616 / 1.01 | 0.536 / 1.13 | 0.754 / 0.80 | 0.872 / 0.52 |
| | LIMSSR (This Paper) | ✓ | 0.830 / 0.60 | 0.885 / 0.52 | 0.636 / 0.87 | 0.826 / 0.60 | 0.595 / 1.07 | 0.606 / 0.91 | 0.755 / 0.76 | 0.885 / 0.52 |
| | $\Delta_{SOTA}$ | - | ↑0.2% / ↑1.7% | ↑1.6% / ↓0.0% | ↓9.5% / ↓7.4% | ↓0.5% / ↑1.7% | ↓3.4% / ↑5.9% | ↑13.1% / ↓19.5% | ↑0.1% / ↓5.0% | ↑1.5% / ↓0.0% |
| CO | ♣ActionMAE (AAAI'23) | ✗ | 0.768 / 0.45 | 0.805 / 0.44 | 0.442 / 0.88 | 0.770 / 0.46 | 0.397 / 1.13 | 0.412 / 1.03 | 0.633 / 0.71 | 0.803 / 0.49 |
| | ♠GCNet (TPAMI'23) | ✗ | 0.729 / 0.47 | 0.775 / 0.48 | 0.486 / 1.25 | 0.729 / 0.49 | 0.279 / 1.50 | 0.361 / 1.64 | 0.592 / 0.97 | 0.764 / 0.58 |
| | ♠IMDer (NeurIPS'23) | ✗ | 0.745 / 0.58 | 0.782 / 0.43 | 0.440 / 0.91 | 0.753 / 0.67 | 0.288 / 1.14 | 0.408 / 0.95 | 0.604 / 0.78 | 0.774 / 0.45 |
| | ♡MLP-Mixer (AAAI'23) | ✗ | 0.732 / 0.72 | 0.660 / 1.96 | 0.452 / 2.93 | 0.779 / 1.03 | 0.511 / 1.24 | 0.153 / 2.18 | 0.580 / 1.68 | 0.784 / 0.36 |
| | ♡PAMFN (TIP'24) | ✗ | 0.802 / 0.57 | 0.763 / 1.42 | 0.606 / 1.68 | 0.773 / 1.39 | 0.506 / 1.81 | 0.233 / 1.40 | 0.648 / 1.38 | 0.804 / 0.53 |
| | ♠MoMKE (ACMMM'24) | ✗ | 0.821 / 0.48 | 0.841 / 0.45 | 0.439 / 0.93 | 0.819 / 0.54 | 0.312 / 1.17 | 0.433 / 0.95 | 0.664 / 0.75 | 0.841 / 0.39 |
| | ♠SDR-GNN (KBS'25) | ✗ | 0.758 / 0.45 | 0.798 / 0.35 | 0.515 / 1.03 | 0.769 / 0.51 | 0.401 / 1.21 | 0.389 / 0.93 | 0.636 / 0.75 | 0.789 / 0.55 |
| | ♡MCMoE (AAAI'26) | ✗ | 0.850 / 0.31 | 0.887 / 0.26 | 0.710 / 0.81 | 0.848 / 0.34 | 0.610 / 1.03 | 0.575 / 0.85 | 0.773 / 0.60 | 0.886 / 0.26 |
| | LIMSSR (This Paper) | ✓ | 0.832 / 0.36 | 0.891 / 0.24 | 0.636 / 0.59 | 0.831 / 0.36 | 0.512 / 1.28 | 0.669 / 0.65 | 0.764 / 0.59 | 0.889 / 0.28 |
| | $\Delta_{SOTA}$ | - | ↓2.1% / ↑16.1% | ↑0.5% / ↑7.7% | ↓3.0% / ↓27.2% | ↓2.0% / ↑5.9% | ↓16.1% / ↑24.3% | ↑16.3% / ↓23.5% | ↓1.2% / ↓1.7% | ↑0.3% / ↑7.7% |
| IN | ♣ActionMAE (AAAI'23) | ✗ | 0.783 / 0.57 | 0.795 / 0.43 | 0.461 / 0.87 | 0.779 / 0.49 | 0.286 / 1.13 | 0.446 / 0.92 | 0.630 / 0.73 | 0.798 / 0.45 |
| | ♠GCNet (TPAMI'23) | ✗ | 0.709 / 1.33 | 0.777 / 0.59 | 0.408 / 2.44 | 0.557 / 4.18 | 0.442 / 1.57 | 0.337 / 2.85 | 0.561 / 2.16 | 0.787 / 0.53 |
| | ♠IMDer (NeurIPS'23) | ✗ | 0.751 / 0.74 | 0.690 / 1.10 | 0.493 / 0.93 | 0.674 / 1.28 | 0.304 / 0.96 | 0.414 / 1.76 | 0.576 / 1.13 | 0.776 / 0.60 |
| | ♡MLP-Mixer (AAAI'23) | ✗ | 0.766 / 0.96 | 0.769 / 1.12 | 0.459 / 3.31 | 0.790 / 0.65 | 0.332 / 2.88 | 0.272 / 3.02 | 0.606 / 1.99 | 0.809 / 0.70 |
| | ♡PAMFN (TIP'24) | ✗ | 0.721 / 2.09 | 0.780 / 0.65 | 0.561 / 0.80 | 0.600 / 3.57 | 0.606 / 2.97 | 0.077 / 3.51 | 0.590 / 2.27 | 0.790 / 0.64 |
| | ♠MoMKE (ACMMM'24) | ✗ | 0.784 / 0.66 | 0.810 / 0.53 | 0.485 / 0.96 | 0.783 / 0.67 | 0.327 / 1.28 | 0.479 / 0.97 | 0.648 / 0.84 | 0.811 / 0.51 |
| | ♠SDR-GNN (KBS'25) | ✗ | 0.732 / 0.86 | 0.800 / 0.57 | 0.508 / 1.09 | 0.685 / 0.79 | 0.458 / 1.08 | 0.440 / 1.27 | 0.625 / 0.94 | 0.822 / 0.43 |
| | ♡MCMoE (AAAI'26) | ✗ | 0.823 / 0.37 | 0.877 / 0.29 | 0.724 / 0.61 | 0.831 / 0.35 | 0.602 / 0.74 | 0.562 / 0.80 | 0.759 / 0.53 | 0.874 / 0.29 |
| | LIMSSR (This Paper) | ✓ | 0.844 / 0.35 | 0.881 / 0.30 | 0.650 / 0.64 | 0.842 / 0.36 | 0.593 / 1.26 | 0.632 / 0.67 | 0.765 / 0.60 | 0.884 / 0.30 |
| | $\Delta_{SOTA}$ | - | ↑2.6% / ↓5.4% | ↑0.5% / ↑3.4% | ↓10.2% / ↑4.9% | ↑1.3% / ↑2.9% | ↓2.1% / ↑70.3% | ↑12.5% / ↓16.2% | ↑0.8% / ↑13.2% | ↑1.1% / ↑3.4% |

*Table 13.* Comparisons of performance on the Fis-V under incomplete multimodal scenarios. $v$, $f$, and $a$ refer to the RGB, flow, and audio modalities. "Average" denotes the average result of all six incomplete multimodal combinations. The **bold** / underline indicate the best / second-best results. ♡, ♣, and ♠ mean the evaluated method sources for incomplete/complete multimodal AQA, incomplete multimodal action recognition, and incomplete multimodal emotion recognition. **T-Miss**: Training-time missing modality. $\Delta_{SOTA}$ means the performance increase or decrease of our LIMSSR compared to the best competing methods.

| Types | Methods | T-Miss | Testing Condition (Spearman Correlation (↑) / Mean Squared Error (↓)) | | | | | | | |
|---|---|---|---|---|---|---|---|---|---|---|
| | | | $\{v,f\}$ | $\{v,a\}$ | $\{f,a\}$ | $\{v\}$ | $\{f\}$ | $\{a\}$ | Average | $\{v,f,a\}$ |
| TES | ♣ActionMAE (AAAI'23) | ✗ | 0.627 / 43.24 | 0.652 / 39.91 | 0.482 / 32.07 | 0.590 / 61.39 | 0.387 / 30.51 | 0.456 / 37.13 | 0.539 / 40.71 | 0.645 / 23.60 |
| | ♠GCNet (TPAMI'23) | ✗ | 0.678 / 19.67 | 0.601 / 28.03 | 0.553 / 27.16 | 0.601 / 27.20 | 0.619 / 24.50 | 0.390 / 43.17 | 0.580 / 28.29 | 0.656 / 21.43 |
| | ♠IMDer (NeurIPS'23) | ✗ | 0.689 / 21.80 | 0.589 / 28.30 | 0.510 / 30.28 | 0.576 / 30.45 | 0.549 / 23.06 | 0.345 / 41.30 | 0.551 / 29.20 | 0.632 / 23.04 |
| | ♡MLP-Mixer (AAAI'23) | ✗ | 0.662 / 52.37 | 0.547 / 81.73 | 0.533 / 38.49 | 0.445 / 86.26 | 0.471 / 40.38 | 0.143 / 113.84 | 0.480 / 68.85 | 0.707 / 20.06 |
| | ♡PAMFN (TIP'24) | ✗ | 0.717 / 58.53 | 0.429 / 79.98 | 0.666 / 169.36 | 0.382 / 140.72 | 0.647 / 162.33 | 0.202 / 89.21 | 0.530 / 116.69 | 0.754 / 22.50 |
| | ♠MoMKE (ACMMM'24) | ✗ | 0.680 / 19.88 | 0.620 / 26.70 | 0.554 / 25.58 | 0.600 / 32.74 | 0.555 / 33.12 | 0.455 / 38.65 | 0.581 / 29.45 | 0.676 / 21.87 |
| | ♠SDR-GNN (KBS'25) | ✗ | 0.678 / 19.01 | 0.618 / 27.21 | 0.569 / 26.26 | 0.619 / 26.41 | 0.579 / 30.84 | 0.417 / 40.16 | 0.585 / 28.31 | 0.672 / 20.83 |
| | ♡MCMoE (AAAI'26) | ✗ | 0.719 / 16.54 | 0.708 / 21.51 | **0.677** / 23.38 | 0.677 / 21.09 | 0.633 / **19.99** | **0.508** / 38.24 | 0.659 / 23.46 | 0.759 / 18.50 |
| | **LIMSSR** (This Paper) | ✓ | **0.774 / 14.64** | **0.758 / 16.58** | 0.703 / **16.61** | **0.703 / 16.61** | **0.686** / 20.46 | 0.486 / 32.71 | **0.684 / 20.57** | **0.792 / 14.82** |
| | $\Delta_{SOTA}$ | - | ↑7.6% / ↓11.5% | ↑7.1% / ↓22.9% | ↓5.6% / ↓4.1% | ↑3.8% / ↓21.2% | ↑6.0% / ↑2.4% | ↓4.3% / ↓11.9% | ↑3.8% / ↓12.3% | ↑4.3% / ↓19.9% |
| PCS | ♣ActionMAE (AAAI'23) | ✗ | 0.768 / 23.84 | 0.702 / 15.32 | 0.655 / 19.04 | 0.641 / 18.75 | 0.570 / 19.22 | 0.514 / 21.44 | 0.649 / 19.60 | 0.745 / 11.07 |
| | ♠GCNet (TPAMI'23) | ✗ | 0.788 / 20.05 | 0.705 / 14.61 | 0.632 / 16.28 | 0.724 / 14.54 | 0.584 / 14.71 | 0.516 / 26.37 | 0.668 / 17.76 | 0.735 / 12.43 |
| | ♠IMDer (NeurIPS'23) | ✗ | 0.797 / 8.58 | 0.718 / 17.01 | 0.621 / 17.70 | 0.755 / 20.44 | 0.679 / 29.71 | 0.462 / 22.63 | 0.685 / 19.34 | 0.762 / 11.00 |
| | ♡MLP-Mixer (AAAI'23) | ✗ | 0.789 / 8.31 | 0.736 / 11.67 | 0.608 / 15.43 | 0.747 / 10.66 | 0.614 / 13.97 | 0.486 / 20.67 | 0.676 / 13.45 | 0.824 / 7.88 |
| | ♡PAMFN (TIP'24) | ✗ | 0.862 / 8.45 | 0.811 / 28.71 | 0.574 / 51.63 | 0.810 / 29.14 | 0.584 / 58.51 | 0.079 / 83.12 | 0.679 / 43.26 | 0.872 / 8.16 |
| | ♠MoMKE (ACMMM'24) | ✗ | 0.813 / 9.81 | 0.747 / 14.50 | 0.723 / 13.66 | 0.753 / 13.57 | 0.734 / 11.81 | 0.537 / 19.54 | 0.727 / 13.81 | 0.805 / 12.74 |
| | ♠SDR-GNN (KBS'25) | ✗ | 0.811 / 10.97 | 0.733 / 14.04 | 0.664 / 15.51 | 0.748 / 14.11 | 0.707 / 12.15 | 0.536 / 24.11 | 0.709 / 15.15 | 0.784 / 12.06 |
| | ♡MCMoE (AAAI'26) | ✗ | **0.878** / 5.49 | 0.847 / 7.76 | **0.770** / 11.44 | 0.831 / 9.19 | **0.753** / 10.79 | 0.603 / 18.84 | **0.795** / 10.58 | **0.880** / 5.61 |
| | **LIMSSR** (This Paper) | ✓ | 0.864 / **5.70** | **0.870 / 6.04** | 0.740 / **9.92** | **0.853 / 7.21** | 0.668 / 13.92 | **0.658** / 14.88 | 0.792 / **9.61** | **0.880 / 5.55** |
| | $\Delta_{SOTA}$ | - | ↓1.6% / ↑3.8% | ↑2.7% / ↓22.2% | ↓3.9% / ↓13.3% | ↑2.6% / ↓21.5% | ↓11.3% / ↑29.0% | ↑9.1% / ↓21.0% | ↓0.4% / ↓9.2% | ↑0.0% / ↓4.5% |

*Table 14.* Comparisons of performance on the RG under incomplete multimodal scenarios. $v$, $f$, and $a$ refer to the RGB, flow, and audio modalities. "Average" denotes the average result of all six incomplete multimodal combinations. The **bold** / underline indicate the best / second-best results. ♡, ♣, and ♠ mean the evaluated method sources for incomplete/complete multimodal AQA, incomplete multimodal action recognition, and incomplete multimodal emotion recognition. **T-Miss**: Training-time missing modality. $\Delta_{SOTA}$ means the performance increase or decrease of our LIMSSR compared to the best competing methods.

| Types | Methods | T-Miss | Testing Condition (Spearman Correlation (↑) / Mean Squared Error (↓)) | | | | | | | |
|---|---|---|---|---|---|---|---|---|---|---|
| | | | $\{v,f\}$ | $\{v,a\}$ | $\{f,a\}$ | $\{v\}$ | $\{f\}$ | $\{a\}$ | Average | $\{v,f,a\}$ |
| Ball | ♣ActionMAE (AAAI'23) | ✗ | 0.650 / 8.13 | 0.554 / 10.30 | 0.530 / 10.50 | 0.615 / 9.91 | 0.471 / 9.45 | 0.367 / 12.37 | 0.537 / 10.11 | 0.613 / 8.77 |
| | ♠GCNet (TPAMI'23) | ✗ | 0.627 / 8.02 | 0.621 / 7.70 | 0.485 / 9.21 | 0.598 / 8.39 | 0.446 / 10.37 | 0.422 / 15.44 | 0.539 / 9.85 | 0.616 / 7.31 |
| | ♠IMDer (NeurIPS'23) | ✗ | 0.646 / 7.80 | 0.596 / 8.27 | 0.527 / 8.36 | 0.623 / 9.10 | 0.532 / 8.53 | 0.333 / 12.33 | 0.550 / 9.06 | 0.654 / 7.30 |
| | ♡MLP-Mixer (AAAI'23) | ✗ | 0.662 / 9.52 | 0.618 / 9.78 | 0.464 / 11.33 | 0.561 / 13.76 | 0.439 / 14.81 | 0.359 / 18.33 | 0.525 / 12.92 | 0.694 / 8.09 |
| | ♡PAMFN (TIP'24) | ✗ | 0.739 / 6.90 | 0.702 / 8.24 | 0.236 / 149.42 | 0.624 / 8.77 | 0.343 / 144.22 | 0.157 / 77.03 | 0.501 / 77.03 | 0.757 / 6.24 |
| | ♠MoMKE (ACMMM'24) | ✗ | 0.652 / 7.74 | 0.592 / 9.09 | 0.603 / **7.30** | 0.524 / 10.75 | 0.564 / 7.83 | **0.442** / 11.75 | 0.566 / 9.08 | 0.670 / 7.47 |
| | ♠SDR-GNN (KBS'25) | ✗ | 0.646 / 6.94 | 0.611 / 8.49 | 0.624 / 8.11 | 0.631 / 9.01 | 0.473 / 8.48 | 0.439 / 12.08 | 0.576 / 8.85 | 0.643 / 7.64 |
| | ♡MCMoE (AAAI'26) | ✗ | **0.803** / 6.05 | 0.732 / 7.45 | **0.643** / 7.96 | 0.719 / 8.28 | **0.639** / 8.22 | 0.327 / 13.19 | **0.664** / 8.28 | 0.806 / 5.66 |
| | **LIMSSR** (This Paper) | ✓ | 0.796 / **5.31** | **0.771 / 6.12** | 0.630 / 7.54 | **0.752 / 6.05** | 0.607 / **7.52** | 0.417 / 12.34 | 0.680 / **7.48** | **0.813 / 5.50** |
| | $\Delta_{SOTA}$ | - | ↓0.9% / ↓12.2% | ↑5.3% / ↓17.9% | ↓2.0% / ↑3.3% | ↑4.6% / ↓26.9% | ↓5.0% / ↓4.0% | ↓5.7% / ↑5.5% | ↑2.4% / ↓9.7% | ↑0.9% / ↓2.8% |
| Clubs | ♣ActionMAE (AAAI'23) | ✗ | 0.778 / 5.25 | 0.590 / 8.36 | 0.433 / 9.43 | 0.734 / 16.29 | 0.436 / 14.37 | 0.117 / **8.49** | 0.549 / 10.37 | 0.668 / 6.35 |
| | ♠GCNet (TPAMI'23) | ✗ | 0.781 / 5.51 | 0.628 / 7.36 | 0.477 / 10.43 | 0.729 / 6.36 | 0.622 / 8.68 | 0.026 / 11.96 | 0.581 / 8.38 | 0.709 / 7.11 |
| | ♠IMDer (NeurIPS'23) | ✗ | 0.792 / 5.43 | 0.615 / 7.61 | 0.405 / 9.66 | 0.729 / 5.43 | 0.634 / 8.05 | 0.069 / 12.34 | 0.579 / 8.09 | 0.698 / 5.80 |
| | ♡MLP-Mixer (AAAI'23) | ✗ | 0.768 / 4.96 | 0.468 / 9.86 | 0.331 / 10.91 | 0.714 / 5.96 | 0.599 / 7.15 | 0.131 / 17.82 | 0.535 / 9.44 | 0.736 / 6.13 |
| | ♡PAMFN (TIP'24) | ✗ | 0.747 / 5.36 | 0.582 / 8.48 | 0.362 / 115.32 | 0.703 / 8.01 | 0.519 / 127.63 | **0.320** / 118.72 | 0.559 / 63.92 | 0.825 / 7.45 |
| | ♠MoMKE (ACMMM'24) | ✗ | 0.783 / 4.65 | 0.589 / 8.14 | 0.509 / 11.23 | 0.731 / 7.12 | 0.654 / 10.05 | 0.056 / 12.31 | 0.589 / 8.92 | 0.709 / 5.91 |
| | ♠SDR-GNN (KBS'25) | ✗ | 0.797 / 4.47 | 0.647 / 7.14 | 0.492 / 10.13 | 0.751 / 6.07 | 0.641 / 8.34 | 0.047 / 12.62 | 0.602 / 8.13 | 0.730 / 6.60 |
| | ♡MCMoE (AAAI'26) | ✗ | 0.820 / **4.10** | 0.720 / **5.58** | 0.633 / 7.22 | **0.773 / 4.85** | 0.657 / 6.82 | 0.005 / 14.25 | 0.648 / 7.14 | 0.815 / **4.22** |
| | **LIMSSR** (This Paper) | ✓ | **0.828** / 4.53 | **0.722** / 6.04 | **0.718 / 5.61** | 0.732 / 5.97 | **0.727 / 5.41** | 0.291 / 13.83 | **0.696 / 6.90** | **0.830** / 4.69 |
| | $\Delta_{SOTA}$ | - | ↑1.0% / ↑10.5% | ↑0.3% / ↑8.2% | ↑13.4% / ↓22.3% | ↓5.3% / ↑23.1% | ↑10.7% / ↓20.7% | ↓9.1% / ↑62.9% | ↑7.4% / ↓3.4% | ↑0.6% / ↑11.1% |
| Hoop | ♣ActionMAE (AAAI'23) | ✗ | 0.737 / 5.64 | 0.692 / 7.02 | 0.595 / 12.44 | 0.736 / 7.28 | 0.504 / 11.53 | 0.202 / 33.18 | 0.602 / 12.85 | 0.771 / 6.25 |
| | ♠GCNet (TPAMI'23) | ✗ | 0.734 / 6.75 | 0.646 / 7.61 | 0.549 / 19.14 | 0.756 / 10.16 | 0.520 / 115.81 | 0.222 / 17.82 | 0.595 / 29.55 | 0.732 / 5.93 |
| | ♠IMDer (NeurIPS'23) | ✗ | 0.769 / **4.31** | 0.690 / 6.99 | 0.517 / 9.84 | 0.761 / 8.38 | 0.531 / 11.93 | 0.333 / 18.98 | 0.622 / 10.07 | 0.754 / 6.41 |
| | ♡MLP-Mixer (AAAI'23) | ✗ | 0.701 / 8.08 | 0.682 / 27.60 | 0.468 / 9.84 | 0.641 / 7.35 | 0.525 / 13.97 | 0.122 / 13.29 | 0.546 / 13.35 | 0.787 / 7.68 |
| | ♡PAMFN (TIP'24) | ✗ | 0.767 / 5.42 | 0.733 / 10.08 | 0.524 / 201.27 | 0.702 / 7.53 | 0.600 / 202.95 | 0.042 / 204.06 | 0.598 / 105.22 | 0.836 / **5.21** |
| | ♠MoMKE (ACMMM'24) | ✗ | 0.780 / 4.93 | 0.756 / 6.60 | **0.602** / 10.12 | 0.779 / 7.34 | 0.552 / 12.83 | 0.354 / 13.36 | 0.661 / 9.20 | 0.789 / 6.01 |
| | ♠SDR-GNN (KBS'25) | ✗ | 0.765 / 6.85 | 0.681 / 7.02 | 0.572 / 11.44 | 0.778 / 8.68 | 0.541 / 13.49 | 0.331 / 14.34 | 0.633 / 10.30 | 0.763 / 5.76 |
| | ♡MCMoE (AAAI'26) | ✗ | 0.796 / 6.59 | 0.809 / 5.66 | **0.771 / 8.62** | 0.757 / 6.53 | **0.643** / 10.53 | **0.490** / 12.06 | **0.726** / 8.33 | 0.845 / 5.62 |
| | **LIMSSR** (This Paper) | ✓ | **0.837** / 4.78 | **0.836 / 5.44** | 0.592 / 8.88 | **0.815 / 5.54** | 0.657 / **7.51** | 0.365 / **11.83** | 0.717 / **7.33** | **0.850** / 5.33 |
| | $\Delta_{SOTA}$ | - | ↑5.2% / ↑10.9% | ↑3.3% / ↓3.9% | ↓23.2% / ↑3.0% | ↑4.6% / ↓15.2% | ↑2.2% / ↓28.7% | ↓25.5% / ↓1.9% | ↓1.2% / ↓12.0% | ↑0.6% / ↑2.3% |
| Ribbon | ♣ActionMAE (AAAI'23) | ✗ | 0.719 / 10.17 | 0.635 / 9.37 | 0.609 / 9.18 | 0.658 / 16.15 | 0.651 / 9.79 | 0.309 / 13.31 | 0.609 / 11.33 | 0.759 / 6.69 |
| | ♠GCNet (TPAMI'23) | ✗ | 0.785 / 6.46 | 0.656 / 9.13 | 0.688 / 8.22 | 0.702 / 7.55 | 0.660 / 9.18 | 0.210 / 15.57 | 0.642 / 9.35 | 0.786 / 5.46 |
| | ♠IMDer (NeurIPS'23) | ✗ | 0.757 / 6.57 | 0.676 / 7.34 | 0.759 / **7.33** | 0.667 / 7.45 | 0.673 / 8.88 | 0.073 / 13.12 | 0.635 / 8.45 | 0.776 / 5.96 |
| | ♡MLP-Mixer (AAAI'23) | ✗ | 0.787 / 6.37 | 0.663 / 9.11 | 0.644 / 8.65 | 0.690 / 11.62 | 0.676 / 9.88 | 0.350 / 14.61 | 0.651 / 10.04 | 0.789 / 8.03 |
| | ♡PAMFN (TIP'24) | ✗ | 0.798 / 9.42 | 0.381 / 128.67 | 0.622 / 22.44 | 0.594 / 135.13 | 0.449 / 18.95 | -0.004 / 139.33 | 0.511 / 75.66 | 0.846 / 7.67 |
| | ♠MoMKE (ACMMM'24) | ✗ | 0.809 / 5.42 | 0.662 / 8.07 | **0.764** / 8.92 | 0.695 / 8.48 | 0.699 / 9.60 | 0.180 / 15.57 | 0.667 / 9.34 | 0.800 / 5.34 |
| | ♠SDR-GNN (KBS'25) | ✗ | 0.801 / 6.07 | 0.676 / 6.86 | 0.729 / 7.91 | 0.730 / 7.33 | 0.686 / 8.91 | 0.217 / 15.07 | 0.668 / 8.69 | 0.810 / 5.41 |
| | ♡MCMoE (AAAI'26) | ✗ | **0.861** / 4.57 | 0.841 / 4.64 | 0.730 / 8.80 | 0.811 / 5.33 | **0.706** / 8.79 | 0.256 / 14.78 | 0.741 / 7.82 | 0.890 / **3.89** |
| | **LIMSSR** (This Paper) | ✓ | 0.836 / **4.89** | **0.876 / 4.35** | 0.672 / 8.86 | **0.813 / 4.88** | 0.684 / **8.23** | **0.358** / 13.44 | **0.742** / **7.44** | **0.908** / 4.76 |
| | $\Delta_{SOTA}$ | - | ↓2.9% / ↑7.0% | ↑4.2% / ↓6.3% | ↓12.0% / ↑20.9% | ↑0.2% / ↓8.4% | ↓3.1% / ↓6.4% | ↑2.3% / ↑2.4% | ↑0.1% / ↓4.9% | ↑2.0% / ↑22.4% |

