# OpenReview forum: "LIMSSR: LLM-Driven Sequence-to-Score Reasoning under Training-Time Incomplete Multimodal Observations"
_ICML.cc/2026/Conference — ICML 2026 spotlight_

### Official Review · Reviewer_j3PN · 2026-02-28

**Soundness:** 2
**Presentation:** 3
**Significance:** 2
**Originality:** 3
**Overall Recommendation:** 4
**Confidence:** 2

**Summary:**

Authors introduce a framework for Incomplete Multimodal Learning (IML) that operates without the need for complete data (allows missing modalities for some samples) during training. They formulate this problem as a conditional sequence reasoning task and use an LLM to infer missing modalities from available modalities. Extensive testing on Action Quality Assessment (AQA) datasets shows it achieves new state-of-the-art performance in data-incomplete scenarios. They also visualize and measure the similarity of predicted missing modalities for further validation.

**Compliance With Llm Reviewing Policy:**

Affirmed.

**Key Questions For Authors:**

1. Missingness distribution - what if one modality is more missing than others? How does this affect performance? Is there a minimum percentage of per-modality data required for training?
2. The performance is lower on smaller datasets - can we calculate a lower bound of data quantity needed to perform well?
3. Generated feature realism - can generated features correspond to non-realistic modality data? E.g. can a feature correspond to OF that cannot exist in the real world?

**Strengths And Weaknesses:**

**Strengths**

1. Novel Problem Setting exploring highly practical train-time incomplete observations
2. Efficient latent reconstruction of missing modalities
3. Extensive evaluations, both quantitative and qualitative
4. Efficient setups using small model (<1B) finetuned on single GPU

**Weaknesses**

1. Limited diversity of application: downstream evaluation is limited to AQA
2. Compared to prior work, the model size appears much larger - raises some concerns of fair comparison.
3. Method constrained to only three modalities where two are highly correlated (RGB & Flow)

---

> ### Author Rebuttal · Authors · 2026-03-31
>
> Thanks for your constructive comments! We hope that our response has addressed all your concerns.
> # [W1] Limited Diversity of Application
> Thanks for the valuable comment. To demonstrate generalizability, we evaluate on CMU-MOSI (2,199 clips) for incomplete multimodal sentiment analysis. **Ours outperforms SOTAs by 1% and 0.7% under incomplete and complete settings, respectively, validating broader applicability.** Detailed results are provided in Rev. XidE [W1,Q1].
> # [W2] Model Size Appears Much Larger
> We clarify that the LLM backbone is not used for unfair scaling, but to address challenging **training-time missingness**, a setting beyond prior “full-modality training” assumptions. To this end, we formulate this as a conditional sequence reasoning problem, leveraging the LLM's strong sequence understanding capabilities. To avoid excessive parameters, we strategically adopt an edge-deployable sub-1B model (Qwen3-0.6B) as our backbone. Computational costs and edge deployability are detailed in **Appendix D and Table 10**.
>
> Additionally, under the same model size (0.6B), our method significantly outperforms DisPro (CVPR'25) [1] by **1.2%/15.4% in ρ/MSE**, showing gains stem from our effective module design rather than scale. Please refer to Rev. XALd [W1,W2,Q2,Q3] for details.
> # [W3] Only Three Modalities & Two are Highly Correlated
> We clarify that the trimodal setup is widely adopted in multimodal tasks (e.g., AQA, emotion recognition, sentiment analysis, and action recognition). **To ensure fair comparison with prior works designed for a trimodal scene**, we align with this widely adopted setting. Although RGB and Flow are correlated, they capture complementary information in AQA: RGB for core visual action execution and Flow for fine-grained motion patterns. **Thus, their combination is an essential and mainstream choice in AQA[2,3].** To explore a more distinct modality set, we evaluate the RGB/Skeleton/Audio setting on the Free Skating (PCS Type) subset of the new FineFS dataset[4], which labels skeleton data:
> |ρ↑/MSE↓|Incomplete|Complete
> |-|-|-
> |PAMFN[2]|0.638/104.42|0.865/82.24
> |MoMKE[5]|0.698/94.73|0.824/93.34
> |MCMoE[3]|0.763/79.15|0.881/73.57
> |**Ours**|**0.779/77.52**|**0.891/71.90**
>
> Our method achieves SOTA results on the RGB/Skeleton/Audio combination, **outperforming MCMoE by an average 1.6%/2.2% in ρ/MSE.** This likely stems from our conditional sequence reasoning paradigm, which naturally adapts to diverse multimodal combinations without relying on specific inter-modality correlations.
> # [Q1] Unbalanced Missingness
> We'd like to clarify that, following [3,5], we study "fixed missing", a more challenging scene than partial missing, where entire modalities are absent. This is inherently a special case of unbalanced missingness (e.g., a missing rate of 0 for RGB and 1 for Flow/Audio). Our method achieves SOTA under such challenging scenes. We further provide new experiments on Fis-V under partial and unbalanced missing:
> |Missing Rate|ρ↑|MSE↓
> |-|-|-
> |0.0|0.841|10.19
> |0.3|0.828|10.83
> |0.5|0.795|12.16
> |0.7|0.756|15.45
> |{v,f}-0.3,a-0.5|0.815|11.17
> |{v,a}-0.3,f-0.5|0.806|11.36
> |{a,f}-0.3,v-0.5|0.799|11.40
>
> Our model remains strong performance when ≥50% of modality data is available. Under unbalanced missing, the RGB (v) has the most significant impact, followed by Flow (f).
> # [Q2] Data Quantity Lower Bound
> Thanks for the good question. We provide new experiments on the largest FS1000-PCS (1000 training samples):
> ||Incomplete|Complete
> |-|-|-
> |100%|0.772/14.12|0.892/6.84
> |50%|0.751/15.78|0.875/7.79
> |25%|0.723/17.84|0.851/9.58
> |MCMoE[3]|0.759/14.97|0.872/7.43
> |MoMKE[5]|0.638/25.54|0.800/17.61
>
> Ours remains robust even with only 50% of the training data. Despite a notable drop at 25%, ours remains competitive. **We estimate that ~250-300 samples are sufficient for acceptable performance.**
> # [Q3] Generated Feature Realism
> Ours infers task-oriented **latent semantics**, not raw pixels/waveforms, so they may not explicitly decode into realistic raw data. However, to prevent physically implausible semantic "hallucinations" (e.g., conflicting motions), our MDA explicitly anchors the LLM's generative reasoning with deterministic feature statistics from available modalities, strictly aligning inferred semantics with real-world contexts. **Appendix Figs. 5-8** show that generated features provide meaningful, task-relevant latent semantics.
>
> ---
> Thanks for your valuable feedback. We will include these discussions in the final version.
> > Reference
>
> [1] Distilled Prompt Learning for Incomplete Multimodal Survival Prediction. CVPR 2025.
>
> [2] Multimodal Action Quality Assessment. TIP 2024.
>
> [3] MCMoE: Completing Missing Modalities with Mixture of Experts for Incomplete Multimodal Action Quality Assessment. AAAI 2026.
>
> [4] Localization-assisted Uncertainty Score Disentanglement Network for Action Quality Assessment. ACMMM 2023.
>
> [5] Leveraging Knowledge of Modality Experts for Incomplete Multimodal Learning. ACMMM 2024.

---

> > ### Author Rebuttal · Reviewer_j3PN · 2026-04-02
> >
> > All concerns are addressed. Already rated weak accept for this paper.

---

> > > ### Author Response · Authors · 2026-04-03
> > >
> > > We truly appreciate your kind reassessment and are glad that the concerns have been fully resolved. Thank you for the thoughtful feedback throughout the process; we will faithfully incorporate these improvements to enhance the final quality of our work. We hope these clarifications and updates are helpful for your final evaluation of our manuscript. Thank you again for the time you devoted to evaluating our work.

---

### Official Review · Reviewer_idHR · 2026-03-11

**Soundness:** 3
**Presentation:** 4
**Significance:** 3
**Originality:** 4
**Overall Recommendation:** 4
**Confidence:** 2

**Summary:**

The paper introduces LIMSSR, an LLM-driven framework designed for Incomplete Multimodal Learning (IML), specifically instantiated for Action Quality Assessment (AQA) tasks. Existing IML methods predominantly rely on the unrealistic assumption that complete modalities are available during training to provide reconstruction supervision or distillation priors. However, real-world training data often suffers from missing modalities. By reformulating IML as a conditional sequence reasoning task, LIMSSR utilizes the semantic reasoning capabilities of Large Language Models (LLMs) to infer latent representations for missing modalities. The framework incorporates Prompt-Guided Context-Aware Modality Imputation (PCMI) with distinct placeholder tokens , LLM-Driven Multidimensional Representation Fusion (LMRF) , and a Mask-Aware Dual-Path Aggregation (MDA) module to dynamically balance generative semantic reasoning with discriminative feature statistics to prevent hallucinations. Extensive experiments on three AQA datasets demonstrate that the proposed method outperforms state-of-the-art baselines.

**Compliance With Llm Reviewing Policy:**

Affirmed.

**Final Justification:**

The author has addressed some of my concerns, and I will keep my score.

**Key Questions For Authors:**

1）Is this framework effective on other multimodal tasks?

2）Will different prompt expressions or structural changes significantly affect model performance?

3）In the cross-modal pattern recovery of Path 2, temporal pooling is applied before cross-modal attention to obtain modal vectors. Will this pooling operation cause the loss of fine-grained temporal dynamics, which are crucial for long-term AQA tasks?

**Limitations:**

yes

**Strengths And Weaknesses:**

Strengths:

1) This paper has strong practical value, because real-world data collection often suffers from systemic missingness due to sensor failures or privacy constraints , and the proposed framework provides a solution for the problem of incomplete modalities during training.

2) Mask-Aware Dual-Path Aggregation (MDA) mitigates the inherent hallucination risks of LLMs by combining high-level semantic reasoning and low-level cross-modal pattern recovery. The framework conducted sufficient and comprehensive experiments on three benchmark datasets and achieved SOTA results, successfully surpassing baseline methods in both missing and complete modality scenarios.

Weaknesses:

Although the paper has analyzed the role of different Prompt components through ablation experiments , this experiment mainly focuses on the performance changes brought by component removal, without further analyzing the robustness issue of the Prompt design, for example, whether different prompt expressions or structural changes will significantly affect the model performance.

---

> ### Author Rebuttal · Authors · 2026-03-31
>
> Thanks for your constructive comments and for recognizing our strengths! We appreciate your rating of our “Presentation” and “Originality” as “4: Excellent”; this has been a great encouragement to us. We hope that our response below has addressed all of your concerns.
> # [Q1] Effective on Other Multimodal Tasks
> We sincerely appreciate your valuable comments. To demonstrate generalization on other multimodal tasks, we evaluate our method on popular CMU-MOSI (2,199 clips) for incomplete multimodal sentiment analysis. Overall, **our approach outperforms SOTA baselines by 1% and 0.7% under incomplete and complete settings, respectively, validating its broad applicability.** Please refer to our response to Rev. XidE [W1,Q1] for detailed results and discussion.
> # [W1,Q2] Prompt Robustness
> We appreciate your acknowledgment of our ablation study on the roles of different prompt components. We are also grateful for your valuable feedback regarding prompt robustness. In response, we conduct new experiments on Fis-V using ρ↑/MSE↓ to investigate the impact of different prompt structures. We establish the following prompt design controls:
> - Template 1(Original):
> ```
> Given the available {avail.} features from an action video. The {miss.} modality is missing. Based on the available modalities, please infer and reconstruct the useful latent representations for the missing {miss.} modalities at the designated positions. Then integrate and enhance all multimodal features for action quality assessment. Output the fused multi-dimensional feature representations at the designated feature dimension positions:
> ```
> - Template 2 (Structured):
> ```
> Available Modalities: {avail.}
> Missing Modalities: {miss.}
>
> Task:
> 1. Infer the latent representations of the missing modalities based on available inputs.
> 2. Reconstruct the missing modality features at the designated positions.
> 3. Fuse all modalities to enhance feature representations for action quality assessment.
>
> Output:
> Output the fused feature representations at the designated feature dimension positions.
> ```
> - Template 3 (Reasoning-Oriented):
> ```
> Given the available modalities {avail.} and missing modalities {miss.}, reason about the relationships between modalities. First, infer the latent representations of the missing modalities based on cross-modal dependencies. Then reconstruct the missing features and integrate them with available ones. Finally, produce enhanced multimodal representations for action quality assessment.
>
> Output the fused feature representations at the designated feature dimension positions.
> ```
> - Template 4 (Multi-step Explicit):
> ```
> You are given multimodal features from an action video.
>
> Step 1: Identify the available modalities: {avail.} and missing modalities: {miss.}
> Step 2: Estimate the latent representations of the missing modalities.
> Step 3: Reconstruct missing modality features at the designated positions.
> Step 4: Fuse all modalities into a unified representation for action quality assessment.
>
> Output the fused feature representations at the designated feature dimension positions.
> ```
> The results are shown below:
> ||Incomplete|Complete
> |-|-|-
> |**Template 1 (Ours)**|0.743/15.09|0.841/10.19
> |Template 2|0.739/15.23|0.838/10.49
> |Template 3|0.747/14.98|0.844/10.05
> |Template 4|0.741/15.30|0.839/10.32
>
> Across diverse prompt structures (structured, reasoning-style, and multi-step), performance remains highly consistent, **with ≤0.004 variance in ρ and ≤0.3 in MSE.** Reasoning-style prompts show slight gains, while structured and multi-step variants yield comparable results. This demonstrates that our framework is robust to prompt formulation.
> # [Q3] Temporal Pooling vs. Fine-grained Dynamics
> Temporal pooling in Path 2 does compress fine-grained temporal signals, but does not discard them. The pooled vectors are derived from $\mathbf{H}_{out}$, where temporal dynamics have already been encoded by the LLM via sequence modeling. Thus, pooling mainly aggregates high-order temporal patterns rather than raw clip-level details.
>
> This design is intentional: Path 2 targets robust cross-modal recovery under missingness, where pooling reduces noise and modality misalignment. Importantly, fine-grained temporal dynamics are still preserved in Path 1 via LLM-based sequential reasoning and further modeled by $\mathcal{H}_{sem}$.
>
> We also replace pooling with clip-level cross-attention. On Fis-V, results slightly improve to 0.747/14.90 & 0.843/10.10 (vs. 0.743/15.09 & 0.841/10.19), **but introduce +1.65 GFLOPs (vs. 0.57 GFLOPs of all our designs), leading to much higher cost.**
>
> Overall, dual-path collaboration balances long-range temporal modeling (Path 1) and robust modality cues (Path 2) without sacrificing efficiency. Empirically, the consistent performance on three long-term datasets shows that temporal modeling is not compromised.
>
> ---
> Thank you once again for your time and effort. We look forward to incorporating these insightful discussions into the final version.

---

> > ### Author Rebuttal · Reviewer_idHR · 2026-04-03
> >
> > My doubts have been resolved.

---

> > > ### Author Response · Authors · 2026-04-03
> > >
> > > We truly appreciate your kind reassessment and are glad that the concerns have been fully resolved. Thank you for the thoughtful feedback throughout the process; we will faithfully incorporate these improvements to enhance the final quality of our work. We hope these clarifications and updates are helpful for your final evaluation of our manuscript. Thank you again for the time you devoted to evaluating our work.

---

### Official Review · Reviewer_XALd · 2026-03-12

**Soundness:** 3
**Presentation:** 3
**Significance:** 3
**Originality:** 3
**Overall Recommendation:** 5
**Confidence:** 3

**Summary:**

This paper studies **incomplete multimodal learning when modalities can be missing already at training time**, where the model cannot rely on complete data to provide reconstruction supervision or teacher priors. Focusing on **multimodal long-term Action Quality Assessment (AQA)** with video (RGB), optical flow, and audio, the authors propose **LIMSSR**, which reframes the task as a **conditional sequence-to-score reasoning** problem driven by an LLM. The approach uses frozen modality-specific feature extractors, projects each modality’s features into the LLM hidden space, and constructs an input sequence with modality boundary tokens; missing modalities are represented by learnable placeholder embeddings and are explicitly described via task prompts (**prompt-guided context-aware modality imputation**). To fuse multimodal information, LIMSSR appends a set of **fusion tokens** that act as “slots” to aggregate multi-dimensional representations from the preceding context. To improve robustness under severe missingness, it introduces a **mask-aware dual-path aggregation** mechanism that combines an LLM-based semantic reasoning path with a feature-based pattern recovery path, trained with regression loss plus consistency and regularization terms. Experiments on three AQA datasets (FS1000, Fis-V, and Rhythmic Gymnastics) evaluate all modality subsets and report improvements over prior baselines under training-time incomplete observations.

**Compliance With Llm Reviewing Policy:**

Affirmed.

**Final Justification:**

The author has solved most of my problems. I have raised my rating to 5: Accept.

**Key Questions For Authors:**

1. **How can the contribution of the LLM be isolated?**
   The current ablations/comparisons do not exclude the possibility that gains primarily come from LLM pretraining. Can the authors add a controlled baseline that isolates the LLM effect (e.g., replacing the LLM with a non-language-pretrained Transformer/sequence model of comparable parameter count) to quantify “LLM itself vs. PCMI/MDA mechanisms”?

2. **Why are comparable LLM/prompt-based baselines not included?**
   Existing baselines are mostly non-LLM methods, and there is no comparable missing-modality approach that also uses an LLM (e.g., prompt-based strategies). Can the authors include at least one representative LLM/prompt-based baseline (even a simple prompt + modality dropout/missing-token marking strategy), or clearly justify why it cannot be included?

3. **Why not use an MLLM/Omni-MLLM backbone (or at least run a controlled attempt)?**
   Since the method essentially follows an Encoder→Projector→LLM bridging paradigm, an MLLM/Omni-MLLM with stronger native cross-modal alignment may provide a higher upper bound. Can the authors provide a controlled ablation (e.g., keeping Encoder+Projector fixed and swapping in the language core of an MLLM/Omni model; or trying a scale-appropriate VLM/Omni such as Qwen3.5-0.8B and Qwen2.5-Omni-3B), or explain why this is infeasible?

4. **Is the optical-flow missingness setting physically realistic?**
   The paper treats RGB/flow/audio as independently missing/present, but flow is typically derived from RGB. Can the authors clarify how flow is obtained when RGB is missing (independent sensor, precomputed cache, or implicitly dependent on RGB), and whether some implausible missing-modality combinations should be removed or constrained?

5. **What evidence supports generalization beyond AQA?**
   The paper claims the framework generalizes to other IML tasks, but experiments focus on AQA. Can the authors add at least one non-AQA missing-modality classification/regression task (even small-scale) to support the “general framework” claim, or more clearly scope and limit the claim?

---

**If the authors can address these issues, I would be inclined to increase my score.**

**Limitations:**

Yes

**Strengths And Weaknesses:**

## Strengths

1. **Addresses a more realistic IML setting (training-time missing modalities):** The paper explicitly targets the challenging scenario where *modalities are already missing during training*, avoiding the common “full-modal training” assumption used by reconstruction- or distillation/prior-based methods.

2. **Clear and principled reformulation using LLM reasoning:** LIMSSR reframes incomplete multimodal learning as a **conditional sequence-to-score reasoning** problem, leveraging LLM semantic reasoning to infer latent semantics from available context rather than relying on explicit reconstruction supervision.

3. **Well-motivated modular design for missing-modality inference and fusion:** The framework combines **prompt-guided context-aware modality imputation** with an LLM-driven **multidimensional representation fusion** mechanism (e.g., fusion tokens), which provides an intuitive way to capture multiple evaluation dimensions for AQA.

4. **Explicit mechanism to mitigate hallucination/uncertainty under severe missingness:** The proposed **mask-aware dual-path aggregation** is designed to dynamically calibrate inference and improve robustness when modality information is incomplete.

5. **Comprehensive empirical validation on AQA benchmarks:** The paper evaluates LIMSSR on **three AQA datasets** under training-time incomplete observations and reports consistent gains over prior baselines across missing-modality settings.


## Weaknesses

1. **Insufficient attribution and fairness (the effect of introducing an LLM is not isolated):**
   The current comparisons and ablations do not rule out the key explanation that the performance gains mainly come from using an LLM (and its large-scale pretraining). Most baselines are conventional non-LLM sequence/multimodal models, and the paper lacks comparable missing-modality methods that also use LLMs (e.g., prompt-based or retrieval-augmented approaches). Existing ablations (e.g., removing prompts) still keep the same LLM backbone, making it difficult to quantify the contribution of the LLM itself versus the proposed design (e.g., PCMI/MDA).

2. **Backbone choice lacks critical validation (no coverage of MLLM/Omni-MLLM):**
   The method follows a typical “Encoder → Projector → LLM hidden space” bridging paradigm, but the experiments and discussion do not sufficiently explain or validate why backbones with stronger native cross-modal alignment (MLLM/Omni-MLLM) are not considered. Since the paper only swaps among several *text* LLMs, it cannot answer whether performance is limited by backbone alignment capability, and the claim of being “compatible with any pretrained LLM/MLLM” is not well supported.

3. **Questionable physical plausibility of modality combinations:**
   The paper treats RGB/flow/audio as independently missing/present, but optical flow is often derived from RGB. It is unclear whether flow can realistically exist when RGB is missing. Some missing-modality combinations may therefore be physically implausible, which weakens the real-world relevance and external validity of the evaluation.

4. **Strong generalization claim with insufficient evidence:**
   The evaluation is mainly restricted to AQA (three datasets). The paper claims the framework generalizes to other incomplete multimodal learning (IML) tasks, but provides no cross-task or cross-domain validation. As a result, the “general framework” narrative currently looks closer to an AQA-specific solution.

---

> ### Author Rebuttal · Authors · 2026-03-31
>
> Thanks for your constructive comments and for recognizing our strengths! We hope that our response has addressed all your concerns.
> # [W1,Q1] LLM Contribution Isolation
> We sincerely thank the reviewer for this critical observation. To isolate the contribution of LLM, we conduct controlled ablations with ρ↑/MSE↓ on Fis-V:
> ||Incomplete|Complete|Avg. Δρ|Avg. ΔMSE
> |-|-|-|-|-
> |Small Transformer+MLP (Baseline)|0.558/59.79|0.718/18.67|-|-
> |Random LLM (Large Transformer)|0.606/50.23|0.731/16.85|+5% (Model Scale)|-13%
> |Frozen LLM (Qwen3-0.6B)|0.633/43.56|0.740/16.08|+3% (LLM Pretrain)|-9%
> |LLM+LoRA|0.648/37.33|0.744/15.18|+2% (Fine-tune)|-10%
> |**Our LIMSSR (LLM+PCMI+LMRF+MDA)**|0.743/15.09|0.841/10.19|+14% (Our Design)|-46%
> |Random LLM+PCMI+LMRF+MDA|0.680/20.99|0.788/14.81|+10% (vs. Random LLM)|-35%
>
> **Contribution Breakdown**: Averaged across scenes, our LIMSSR improves the Baseline by +0.154 ρ and -26.59 MSE. The total gain decomposes into: Model Scale (20%/21%), LLM Pretrain (12%/14%), Fine-tune (6%/14%), and **Our Design (62%/51%, the largest contributor)**. While LLM scale and world knowledge provide crucial semantic priors for challenging training-time incompleteness, our design drives the primary performance improvements.
>
> **Synergy & Intrinsic Effectiveness**: Replacing the LLM with a randomly initialized Transformer of equal size (last row) significantly degrades performance. This proves our modules are not conventional imputers, but a novel LLM-driven framework that deeply synergizes with LLM reasoning. Nonetheless, **applying our modules to a "Random LLM" still yields +10%/35% gains over the pure "Random LLM",** surpassing recent SOTAs (e.g., SDR-GNN (KBS'25), MoMKE (MM'24)). This verifies that our dual-path aggregation independently models strong discriminative representations.
> # [W1,W2,Q2,Q3] Compared to LLM/Prompt-based Methods & Using MLLM
> Thanks for the good question about comparing LLM & MLLM. We provide these experiments on Fis-V below in the table:
> ||T-Miss|Incomplete|Complete
> |-|-|-|-
> |MPLMM (ACL'24) [1] (Prompt-based)|×|0.728/18.62|0.811/14.39
> |DisPro-0.6B (CVPR'25) [2] (LLM-based)|×|0.737/16.58|0.828/13.04
> |**Ours-0.6B**|√|0.743/15.09|0.841/10.19
> |Ours+Qwen3.5-0.8B (MLLM)|√|0.745/15.01|0.844/10.23
> |Ours+Qwen3.5-2B (MLLM)|√|0.749/14.56|0.848/9.97
>
> First, we compare ours with SOTA LLM/Prompt-based methods, using the same Qwen3-0.6B as backbone for [2] to ensure fairness. **Under both complete and incomplete settings, ours consistently achieves superior performance.**
>
> Then, we replace the LLM backbone with SOTA MLLMs. Following prior work, we use the same pre-extracted features for fairness. Under this setting, Qwen3.5-0.8B yields marginal improvements over Qwen3-0.6B (Text LLM). Although MLLMs possess stronger multimodal alignment capabilities, their performance is comparable to that of LLM when only feature sequences are input. While Qwen3.5-2B further boosts performance, it significantly increases computational overhead. Thus, our sub-1B LLM choice optimally balances performance and edge-deployability, while retaining scalability.
> # [W3,Q4] Flow Physical Plausibility
> Thanks for the question. We'd like to clarify that all features are **pre-extracted and stored independently**, as detailed in **Appendix C** and in line with prior work (MCMoE-AAAI'26 [3], PAMFN-TIP'24 [4]). **Thus, Flow can exist when RGB is missing.** This setup simulates real-world multi-sensor systems where modalities are captured separately (e.g., Flow from an event camera). Moreover, the RGB/Flow/Audio combination is mainstream and crucial for AQA [3,4], as it aligns with its core evaluation criteria, such as visual action execution, motion pattern attributes, and action-music synchronization. We also provide experiments on a more distinct modality set (RGB/Skeleton/Audio), please see our response to Rev. j3PN [W3].
> # [W4,Q5] Generalization Beyond AQA
> We sincerely appreciate your valuable comments. To demonstrate generalization beyond AQA, we evaluate our method on popular CMU-MOSI [5] (2,199 clips) for incomplete multimodal sentiment analysis. Overall, **our approach outperforms SOTA baselines by 1% and 0.7% under incomplete and complete settings, respectively, validating its broad applicability.** Please refer to our response to Rev. XidE [W1,Q1] for detailed results and discussion.
>
> ---
> Thank you once again for your time and effort. We look forward to incorporating these insightful discussions into the final version.
> > Reference
>
> [1] Multimodal Prompt Learning with Missing Modalities for Sentiment Analysis and Emotion Recognition. ACL 2024.
>
> [2] Distilled Prompt Learning for Incomplete Multimodal Survival Prediction. CVPR 2025.
>
> [3] MCMoE: Completing Missing Modalities with Mixture of Experts for Incomplete Multimodal Action Quality Assessment. AAAI 2026.
>
> [4] Multimodal Action Quality Assessment. TIP 2024.
>
> [5] MOSI: Multimodal Corpus of Sentiment Intensity and Subjectivity Analysis in Online Opinion Videos. arXiv 2016.

---

> > ### Author Rebuttal · Reviewer_XALd · 2026-04-02
> >
> > The author has answered most of my questions, so I will maintain a high rating.

---

> > > ### Author Response · Authors · 2026-04-03
> > >
> > > We truly appreciate your kind reassessment and are glad that the concerns have been fully resolved. Thank you for the thoughtful and constructive feedback throughout the process; we will faithfully incorporate these improvements to further enhance the final quality of our work.
> > >
> > > We are grateful for your earlier comment that you would be inclined to increase your score if the concerns were adequately addressed. We hope that our revisions and clarifications have met your expectations in this regard.
> > >
> > > We sincerely appreciate the time and effort you have devoted to reviewing our work.

---

### Official Review · Reviewer_XidE · 2026-03-12

**Soundness:** 3
**Presentation:** 3
**Significance:** 3
**Originality:** 3
**Overall Recommendation:** 4
**Confidence:** 3

**Summary:**

This submission aims to address a fundamental challenge in incomplete multimodal learning, where modalities may be missing not only at inference time but also during training. The authors intend to examine a notable aspect of multimodal reasoning by reformulating the problem as a conditional sequence-to-score reasoning task using a large language model (LLM). Experiments are conducted on three Action Quality Assessment benchmarks , where the proposed method demonstrates improved or competitive performance compared with existing methods under various incomplete modality settings.

**Compliance With Llm Reviewing Policy:**

Affirmed.

**Final Justification:**

The response basically solves my concerns. I have adjusted the rating and confidence.

**Key Questions For Authors:**

1. Can the proposed LIMSSR framework be applied to other multimodal learning tasks beyond Action Quality Assessment?
2. What is the computational overhead of using an LLM backbone in the proposed framework, especially for long video sequences?
3. How does the model performance change under more extreme missing modality scenarios?

**Limitations:**

yes

**Strengths And Weaknesses:**

# Strength

1. The paper studies a realistic setting where modalities may be missing during both training and inference, which is less explored compared with conventional incomplete multimodal learning setups.
2. The LIMSSR framework integrates prompt-guided modality representation, LLM reasoning, and a mask-aware aggregation module in a relatively structured pipeline.

# Weakness

1. Experiments are restricted to Action Quality Assessment tasks. It remains unclear whether the proposed framework generalizes well to other multimodal learning scenarios such as multimodal classification, emotion recognition, or video understanding.
2. Although the proposed method achieves better results than existing baselines, the performance gains are relatively small, which somewhat limits the practical impact of the proposed approach.

---

> ### Author Rebuttal · Authors · 2026-03-31
>
> Thanks for your constructive comments! We hope that our response has addressed all your concerns.
> # [W1,Q1] Generalization Beyond AQA
> Thanks for the good question. We evaluate ours on popular CMU-MOSI [1] (2,199 clips) for incomplete multimodal sentiment analysis. Following [2], we use audio (a), text (t), and visual (v) modalities, testing under fixed missing and complete settings. We report Acc(%)↑/F1(%)↑ and compare with SOTA methods below:
> ||**T-Miss**\*|a|t|v|a,v|a,t|t,v|**Average***|**a,t,v**
> |-|-|-|-|-|-|-|-|-|-
> |GCNet [3]|×|56.1/54.5|83.7/83.6|56.1/55.7|62.0/61.9|84.5/84.4|84.3/84.2|71.1/70.7|85.2/85.1
> |IMDer [4]|×|62.0/62.2|84.8/84.7|61.3/60.8|63.6/63.4|85.4/85.3|85.5/85.4|73.8/73.6|85.7/85.6
> |MoMKE [2]|×|63.2/58.6|86.6/86.5|63.4/63.3|64.0/64.7|87.2/87.2|87.0/87.0|75.2/74.6|88.0/87.9
> |MCMoE† [5]|×|59.9/59.6|86.3/86.3|63.9/63.5|64.2/64.2|86.9/86.9|86.5/86.4|74.6/74.5|87.4/87.3
> |**Ours**|√|62.8/61.0|86.9/86.9|64.9/64.0|65.5/65.1|87.8/87.8|87.5/87.4|**75.9/75.4**|**88.5/88.5**
>
> \* **T-Miss**: Training-time missing modality. **Average**: average result over all six incomplete settings. † Our reimplementation based on official codes.
>
> The results show that ours achieves superior or comparable performance across all setups. Compared to the SOTA MoMKE [2], it yields **average gains of 1% (Average) and 0.7% (a,t,v)**, despite operating under the more challenging T-Miss setting. This validates its applicability beyond AQA and the efficacy of framing incomplete multimodal learning as a conditional sequence reasoning task.
> # [W2] Gains are Relatively Small
> We appreciate that all reviewers recognize our SOTA results. Regarding your concern about small gains, we emphasize that ours tackles the far more challenging **training-time missingness**, unlike baselines assuming full-modality training. Despite this disadvantage, ours consistently outperforms prior methods. Under incomplete settings, we improve ρ/MSE by **0.9%/8.4%, 1.2%/11.3%, and 1.9%/7.6%** on FS1000, Fis-V, and RG. Under complete settings, without full-modality training, ours still achieves average gains of **1.3%/7.0%** across all datasets. We believe these improvements are substantial, given the strong performance saturation in AQA.
> # [Q2] Computational Overhead
> We'd like to clarify that the computational overhead comparison is already included in **Appendix Table 10**. Excluding the LLM backbone, ours introduces **only 17.35M parameters and 0.57 GFLOPs**, keeping it on par with existing methods. While we utilize an LLM backbone, we strategically select an edge-deployable sub-1B model (Qwen3-0.6B) to ensure computational feasibility. Moreover, by employing LoRA, only 177.26M parameters are updated during training. **These strategies allow ours to be easily trained on a single RTX 3090 GPU, similar to baselines.**
>
> Regarding long video sequences, following [2,3,5], we use pre-extracted multimodal feature sequences rather than feeding raw audio/video into the LLM. For instance, on the FS1000, a sample (95×768) is projected to 95 tokens per modality for LLM. Thus, excluding the fixed prompt, processing all three modalities along with special tokens for a single sample requires **only about 95×3=285 tokens**, which incurs a minimal computational cost.
> # [Q3] Performance under More Extreme Missing
> Thanks for the question. Following [2,4,5], we already evaluated the highly challenging “fixed missing” setting, where entire modalities are completely unavailable. This is already an extreme scene; e.g., inputting only the RGB modality into a trimodal model severely disrupts intrinsic multimodal interactions compared to partial missing.
>
> To explore an even more extreme scene, **we simulate partial missing within a single modality (RGB) under missing rates of 0.1/0.3/0.5** on Fis-V. For TES, performance (ρ↑/MSE↓) drops from 0.703/16.61 (0 missing rate) to 0.700/16.86 (0.1), 0.671/17.74 (0.3), and 0.623/21.85 (0.5). Similarly, for PCS, it drops from 0.853/7.21 to 0.846/7.44, 0.813/8.41, and 0.750/11.76. **Our method maintains robust results at 0.1 and 0.3 missing rates**, with noticeable degradation at the extreme 0.5 rate. Such severe compound missingness is rarely explored and presents a valuable direction for future work.
>
> ---
> **We sincerely hope these comprehensive new results address your concerns and justify a higher rating.** Thank you once again for your time and effort. We will include these insightful discussions in the final version.
> > Reference
>
> [1] MOSI: Multimodal Corpus of Sentiment Intensity and Subjectivity Analysis in Online Opinion Videos. arXiv 2016.
>
> [2] Leveraging Knowledge of Modality Experts for Incomplete Multimodal Learning. ACMMM 2024.
>
> [3] GCNet: Graph Completion Network for Incomplete Multimodal Learning in Conversation. TPAMI 2023.
>
> [4] Incomplete Multimodality-Diffused Emotion Recognition. NeurIPS 2023.
>
> [5] MCMoE: Completing Missing Modalities with Mixture of Experts for Incomplete Multimodal Action Quality Assessment. AAAI 2026.

---

> > ### Author Rebuttal · Reviewer_XidE · 2026-04-05
> >
> > The response basically solves my concerns. I have adjusted the rating and confidence.

---

> > > ### Author Response · Authors · 2026-04-06
> > >
> > > We are very glad to hear that our response has basically resolved your concerns. Thank you for adjusting the rating and confidence, and for your recognition of our work. Should you have any further questions or suggestions, please do not hesitate to let us know.
> > >
> > > We truly appreciate your time and thoughtful feedback throughout the review process. We will faithfully incorporate these improvements to further enhance the final quality of our work.

---

### Decision · Program_Chairs · 2026-04-30

**Decision:**

Accept (spotlight)

**Comment:**

In this paper, the authors proposed a framework for Incomplete Multimodal Learning (IML) that operates without the need for complete data (allows missing modalities for some samples) during training. Originally, reviewers have concerns such as the generalization of the method, insufficient evaluation, etc. After the rebuttal, all reviewers said that their concerns were well resolved by the authors, and recommended acceptance of the paper. The AC carefully read the paper, the rebuttal, and the reviewer discussions, and think the paper has good contrubtion to the community; and thus recommends acceptance of the paper.